# MolPILE - large-scale, diverse dataset for molecular representation learning

## Abstract

The size, diversity, and quality of pretraining datasets critically determine the generalization ability of foundation models. Despite their growing importance in chemoinformatics, the effectiveness of molecular representation learning has been hindered by limitations in existing small molecule datasets. To address this gap, we present MolPILE, large-scale, diverse, and rigorously curated collection of 222 million compounds, constructed from 6 large-scale databases using an automated curation pipeline. We present a comprehensive analysis of current pretraining datasets, highlighting considerable shortcomings for training ML models, and demonstrate how retraining existing models on MolPILE yields improvements in generalization performance. This work provides a standardized resource for model training, addressing the pressing need for an ImageNet-like dataset in molecular chemistry.

## 1 Introduction

Modern chemoinformatics relies extensively on machine learning (ML) methods, particularly for virtual screening and ADMET workflows. In practice, however, drug design problems are often constrained to very small datasets, typically comprising only hundreds or a few thousand molecules (van Tilborg et al., 2024). This presents a major limitation for traditional workflows based on feature extraction methods, such as molecular fingerprints (Adamczyk & Ludynia, 2024), which are subsequently paired with tabular classifiers. Molecular representation learning has emerged as a powerful strategy for incorporating general chemical knowledge from large-scale molecular databases. The transfer learning capabilities of pretrained neural networks mitigate data scarcity either through fine-tuning for specific targets or by employing embeddings from these models as pretrained feature extractors (Praski et al., 2025). The latter approach is particularly important for unsupervised tasks common in chemoinformatics, including molecular clustering (Butina, 1999) and similarity searching (Stumpfe & Bajorath, 2011).

The amount, diversity, and quality of the pretraining dataset directly influence the transfer learning capabilities of models (Gadre et al., 2023; Gao et al., 2020; Abdin et al., 2024). This is particularly critical in chemistry, which encompasses a wide range of subdomains and applications, extending beyond medicinal chemistry (itself defined by numerous ADMET properties) to, e.g., natural products (Ertl et al., 2008), agrochemistry and ecotoxicology (Adamczyk et al., 2025a), materials science and industrial chemistry (Pilania et al., 2013), or food and flavor chemistry (Kou et al., 2023). These domains differ substantially in typical structures, functional groups, and element distributions. Pretrained models must therefore perform robustly not only for organic molecules but also for metals, organometallics, salts, and other classes relevant to established QSAR workflows (Young et al., 2008). Equally important is data quality: pretraining sets should reflect realistic experimental conditions through rigorous deduplication, standardized chemical structures, molecular weight ranges appropriate for small molecules (excluding, e.g., proteins, peptides, or nucleic acids), and constraints on properties such as synthetic accessibility and solubility (e.g., avoiding compounds with extreme logP values).

Deficiencies in existing datasets in the aforementioned aspects have significant implications for ML models. A recent study of Praski et al. (2025) reported that nearly all existing pretrained neural models perform statistically worse than the simple ECFP molecular fingerprint (Rogers & Hahn, 2010) on a large-scale benchmark. Notably, the majority of these models were pretrained on very

limited subsets of ZINC or PubChem, without assuring diversity or quality filtering (see Appendix A for a detailed overview), which may represent a key factor underlying this outcome. Similarly, in Sultan et al. (2024) authors point out the importance of chemical space coverage and data filtering for pretraining molecular models, and note that the disparity in pretraining datasets makes it considerably harder to fairly compare the generalization of models. Their performance differences may not stem from algorithmic advances, but rather pretraining data used. This situation shows the need for a single, high-quality dataset for pretraining molecular ML models.

**Key contributions** of this work are as follows. Primarily, we introduce the **MolPILE dataset**, a large-scale collection of small compounds for molecular representation learning and neural model pretraining, designed to satisfy the key desiderata of size, diversity, and quality. With 222 million molecules, it is the largest publicly available dataset of experimentally verified, synthesizable compounds. It spans a broad chemical space, encompassing substantial variation in elements, structures, and properties. A multi-step processing and filtering workflow further ensures data quality through deduplication, structure standardization, and real-world feasibility filtering. In essence, it is intended to serve a role for molecular machine learning comparable to that of ImageNet in computer vision and PILE in natural language processing, providing a unified and standardized pretraining dataset.

In addition, we provide a **comprehensive examination of the chemical spaces** represented in commonly used pretraining datasets, including UniChem (Chambers et al., 2013), PubChem (Kim et al., 2015), and ZINC (Sterling & Irwin, 2015). Comparative analyses demonstrate the superior diversity and quality of MolPILE, while also identifying specific deficiencies in existing datasets.

Finally, we show how these dataset characteristics **translate directly into model performance**. Models such as Mol2vec (Jaeger et al., 2018) and ChemBERTa (Ahmad et al., 2022), when trained on MolPILE, achieve results surpassing those of their original counterparts.

All code and datasets will be released publicly upon submission acceptance, under permissive licenses. Code is available in the supplementary material, following ICLR requirements.

## 2 LITERATURE REVIEW

**Molecular databases creation.** Large-scale small molecule collections can be constructed primarily in three ways: experimentally, by combinatorial enumeration, or with combinatorial reactions.

*Experimental databases* are the most established resources, containing molecules that have been synthesized and tested for specific purposes, such as high-throughput screening (HTS). Their data typically originates from scientific literature, patents, and commercial suppliers. Large aggregate repositories include PubChem (Kim et al., 2015), UniChem (Chambers et al., 2013), and ZINC15 (Sterling & Irwin, 2015), and there also exist more specialized examples, such as COCONUT (Sorokina et al., 2021) and SuperNatural3 (Gallo et al., 2022) for natural products. These collections are generally diverse and realistic, reflecting contributions from a wide range of projects and subfields in chemistry. At present, experimental databases encompass at most roughly 200 million small molecules, though data quality can be inconsistent. For instance, vendors sometimes submit larger biomolecules such as peptides or RNAs, or molecules not parseable by RDKit.

*Combinatorially enumerated* databases are constructed by generating all theoretically possible atomic structures and bonds up to a given size, followed by filtering out chemically implausible arrangements based on predefined rules. A prominent example is GDB-17 (Ruddigkeit et al., 2012). While such databases can reach immense sizes, their diversity, quality, and synthesizability are constrained by the rules applied during generation and filtering, which are difficult to enforce consistently across the entire dataset (Hoffmann & Gastreich, 2019). Despite their scale, these databases often cover only limited regions of chemical space in terms of elements and scaffolds, and are restricted to relatively small molecules due to combinatorial explosion at higher atom counts.

*Combinatorial reaction* databases generate compounds by combining readily available reagents using well-defined chemical reactions, often described with representations such as SMARTS or LHASA patterns. Examples include ZINC20 (Irwin et al., 2020) and Enamine REAL (Grygorenko et al., 2020). This approach enables the creation of large, synthetically accessible datasets by leveraging well-established reaction pathways. However, the resulting chemical space is inherently constrained, as it primarily reflects common and safe routes for synthesis. Moreover, the quality and

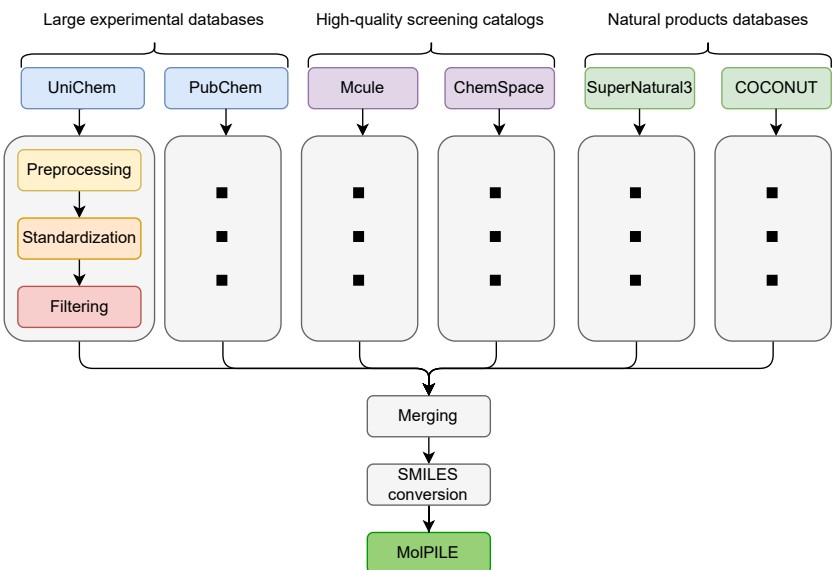

Figure 1: MolPILE data processing workflow.

diversity of the generated compounds depend heavily on the availability and quality of the building blocks used.

**Database filtering.** Many commercial molecular databases apply extensive filtering, such as Lipinski's rule of 5 (Lipinski et al., 2001), which has been criticized for biasing chemical space toward narrow physicochemical ranges (Doak et al., 2016; Walters, 2012; Zhang & Wilkinson, 2007). For instance, Enamine REAL (Grygorenko et al., 2020) applies Lipinski and Veber filters (Veber et al., 2002), Maybridge enforces a variety of structural rules and almost entirely conforms to Lipinski rule of 5 (Maybridge), and ChemDiv uses REOS (Walters & Namchuk, 2003) and PAINS (Baell & Holloway, 2010) filters (ChemDiv). While effective for targeted screening, such preprocessing substantially reduces chemical diversity and limits usefulness for ML pretraining. Broader and more representative coverage is provided by unfiltered sources such as ChemSpace (ChemSpace), Mcule (Kiss et al., 2012), PubChem, and UniChem.

**Pretraining of molecular ML models.** Foundation models benefit from larger parameter counts and datasets, following neural scaling laws (Kaplan et al., 2020), but their performance also depends on the quality and diversity of training data (Ghorbani et al., 2022; Gadre et al., 2023; Sultan et al., 2024). In molecular ML, pretraining datasets are often limited to small subsets of ZINC15 or ChEMBL, typically under 20 million compounds, which cannot fully represent real-world chemical diversity. Examples include SimSon (Lee et al., 2025) (1M) and R-MAT (Maziarka et al., 2024) (4M). At the other extreme, some models train on massive but unfiltered collections, such as ChemFM (Cai et al., 2024) (entire UniChem) and ChemBERTa (Ahmad et al., 2022) (77M subset of PubChem), which include abnormal or non-synthesizable compounds. A third line of work uses subsets of prefiltered databases, or applies additional filtering criteria, which further constrains the available chemical space, as in GEM (Fang et al., 2022) (20M from ZINC) or CDDD (Winter et al., 2019) (six custom filters on ZINC and PubChem). Appendix A provides a detailed comparison.

## 3 MOLPILE DATASET

The main contribution of this work is MolPILE, a large-scale dataset of nearly 222 million small compounds for pretraining molecular ML models. It is created based on multiple sources, with workflow illustrated in Figure 1, with focus on quality and diversity. Each source database was processed using a uniform three-stage pipeline consisting of preprocessing, standardization, and filtering. Resulting datasets were merged and finally deduplicated, yielding the MolPILE dataset.

## 3.1 SOURCE DATA

The foundation of MolPILE is built on UniChem and PubChem, two large general-purpose chemical databases comprising experimentally synthesized compounds. These databases aggregate data from multiple suppliers and other databases; for example, UniChem includes the entirety of ChEMBL (Zdrazil et al., 2023), providing MolPILE with broad coverage of chemical sources. UniChem applies stringent input filtering rules based on the InChI format (Heller et al., 2013), whereas PubChem accepts a wider range of input data and vendors. See Appendix B for an overview of popular sources included in these databases.

We also incorporate Mcule (Kiss et al., 2012) and ChemSpace (ChemSpace), screening catalogs containing commercially available molecules. As such, they are by design high-quality, readily synthesizable, and highly diverse. We selected these two sources because, unlike many alternatives, they do not apply restrictive filters such as the Lipinski Rule of 5 (Lipinski et al., 2001). Aggressive filtering can negatively impact the generalization of chemical foundation models, as it drastically limits chemical space coverage, a point widely discussed in drug design literature (Doak et al., 2016; Walters, 2012; Zhang & Wilkinson, 2007).

Finally, we include SuperNatural3 (Gallo et al., 2022) and COCONUT (Sorokina et al., 2021), two large natural product (NP) databases. Interest in NP-based therapeutics is growing due to their potent bioactivity and recent successes in the field, such as Paclitaxel (Atanasov et al., 2021). However, NPs are underrepresented in typical databases, which primarily focus on synthetic medicinal compounds. Incorporating these NP datasets enable MolPILE to more densely sample NP-like chemical space and enhances their representation in pretrained models.

We select only experimental databases without additional filtering. This ensures their diversity and avoids biasing MolPILE towards any specific subset of chemical compounds.

## 3.2 PROCESSING WORKFLOW

The processing workflow is divided into three distinct phases, described below. Each phase removes molecules that cannot be parsed at that step and performs deduplication based on the InChI format. InChI was chosen as the primary representation because multiple studies have demonstrated its advantages in structure canonicalization and unambiguous representation, which are crucial for data deduplication and minimizing structural redundancy (Akhondi et al., 2015; Fanton et al., 2013; Hersey et al., 2015). RDKit is employed as the main molecular processing framework, offering an open-source environment. See Appendix C for implementation and hardware details.

In the **preprocessing** step, each dataset is converted into a Parquet file containing molecules in InChI format. The diversity of input formats and representations presents a major challenge when integrating data from multiple sources. For example, UniChem provides a CSV with InChI strings, whereas ChemSpace is distributed as an SDF file. After preprocessing, all datasets share a consistent format, allowing subsequent steps to be applied uniformly across datasets.

The **standardization** step focuses on unifying molecular graph representations. They can vary across datasets depending on the tools and frameworks used, leading to subtle structural duplicates and differences, for example in functional group representation. Such inconsistencies can introduce noise during model pretraining. To address this, we implement standard sanitization and cleanup procedures in RDKit, as recommended for general QSAR workflows (Young et al., 2008), including kekulization, valence checks, aromaticity modeling, conjugation and hybridization adjustments, hydrogen removal, metal disconnection, functional group normalization, and reionization. Molecules that fail any of these steps are removed. We intentionally avoid aggressive filtering, such as salt removal, since it is not recommended in certain domains, e.g., agrochemistry (Adamczyk et al., 2025a).

Finally, we introduce a novel **molecular feasibility filter** as part of the filtering step. Its purpose is to remove erroneous molecules, such as those with physically unreasonable values of properties like logP or TPSA, multi-fragment complexes, and excessively large compounds, including potential peptides, crystals, or metal clusters. To design the filter, we analyzed the distributions of these properties in standardized datasets and relevant literature (see Appendix D for details). The filter criteria are as follows: molecular fragments $\leq 3$, length of InChI $< 2000$, molecular weight $\leq 2500$, number of atoms $\leq 150$, HBA $\leq 20$, HBD $\leq 15$, logP in range $[-10, 25]$, TPSA $\leq 500$, number

of rotatable bonds $\leq 60$. Those conditions are intentionally lax, aiming only to remove clearly infeasible molecules without unduly restricting chemical space. This feasibility filter can also serve as a general data quality tool for other molecular ML workflows.

### 3.3 Merging and subset selection

After processing all datasets, we merge them. During all steps, we keep a unique ID for each molecule, e.g. CID in PubChem. In the merge step, we incorporate the datasets sequentially, adding molecules from each source in order of decreasing dataset size. This approach preserves full traceability, allowing us to unambiguously link every molecule to its original database, and also ensures no duplicates. Appendix E provides a detailed breakdown of inter-dataset contributions.

Lastly, we convert all molecules from InChI to SMILES format, as SMILES is the representation most widely adopted in chemical language models and is readily parsed by standard chemoinformatics libraries. The final MolPILE dataset comprises nearly **222 million molecules**. It is freely and publicly available; see Appendix F for exact licensing.

To facilitate experimentation and the training of computationally intensive models, we additionally provide diverse subsets containing 1M, 5M, and 10M molecules. These subsets are constructed using an algorithm inspired by diversity-based selection in virtual screening. Full algorithmic details are provided in Appendix G; here we present a brief overview. We employ the MaxMin approximation to maximum diversity picking (Sayle; Ashton et al., 2002), which identifies a subset of $k$ molecules that maximizes the sum of their pairwise Tanimoto distances. The dataset is then clustered by assigning each compound to its nearest selected molecule, as measured by Tanimoto distance. Molecules are subsequently sampled uniformly from each cluster until the desired subset size is obtained.

## 4 Datasets analysis

In this section, we perform a series of analyses for evaluating the size, diversity, and quality of MolPILE and its source datasets. We also compare with alternative datasets, commonly used for pretraining molecular representation models, with results indicating particular deficiencies in each one in at least one of those three areas.

ChEMBL (Zdrazil et al., 2023) contains bioactivity data from screening assays. While it is included in MolPILE through UniChem, we analyze it separately given its frequent use in model pretraining. GDB-17 (Ruddigkeit et al., 2012) is an enumerated dataset comprising compounds with up to 17 atoms of $C$, $N$, $O$, $S$, and halogens; we employ the publicly available representative subset of 50 million molecules. ZINC15 (Sterling & Irwin, 2015) provides experimentally validated small molecules with an emphasis on medicinal chemistry and rigorous filtering. We use its high-quality subset of 13.7 million molecules, with established 3D structures and verified in-stock vendor availability. We choose ZINC15 rather than ZINC20, as it is more commonly used in model pretraining, and the latter is also almost exclusively combinatorial (enumerated).

### 4.1 Dataset size

The final MolPILE dataset contains nearly 222 million molecules. To illustrate the impact of our multi-step workflow, Table 1 summarizes the number of compounds removed at each step (rounded for readability; see Appendix H for exact numbers). MolPILE has about 33M more compounds than its largest source, UniChem. Overall, the filtering step removes the largest fraction of molecules, particularly from the UniChem and PubChem datasets. This is expected, given the large scale and heterogeneous sources of these databases, which include many lower-quality molecules. In total, our processing pipeline removes over 10M molecules. For ChEMBL and GDB-17, our pipeline would remove almost no compounds, indicating their good quality. However, the main problem with ChEMBL is its small size; at just 2.4M molecules, it is not suitable for training any large-scale models, particularly transformer-based. ZINC loses about 1M molecules out of initial 13.7M, which is quite concerning. These results highlight the importance of high-quality, curated datasets like MolPILE, as almost all models are currently pretrained on raw data containing such erroneous structures.

Table 1: Dataset sizes: initial count, molecules removed at each step, and the final count.

| Dataset | Initial count | Preprocessing | Standardization | Filtering | Final dataset |
|---|---|---|---|---|---|
| UniChem | 189M | 0 | -459k | -4.7M | 184M |
| PubChem | 121.4M | -467k | -107k | -4.2M | 116.6M |
| Mcule | 43.6M | -105k | -157 | -13.6k | 43.4M |
| ChemSpace | 7.8M | -78 | -107 | -4.4k | 7.8M |
| SuperNatural3 | 1.2M | -2.9k | -331 | -44k | 1.1M |
| COCONUT | 695k | -8.6k | -154 | -26k | 660k |
| ChEMBL | 2.4M | -8 | -911 | -41k | 2.4M |
| GDB-17 | 50M | -4.3k | -5 | 0 | 50M |
| ZINC | 13.7M | -945k | -8.7k | -3k | 12.7M |
| **MolPILE** | | | | | 222M (221,950,487) |

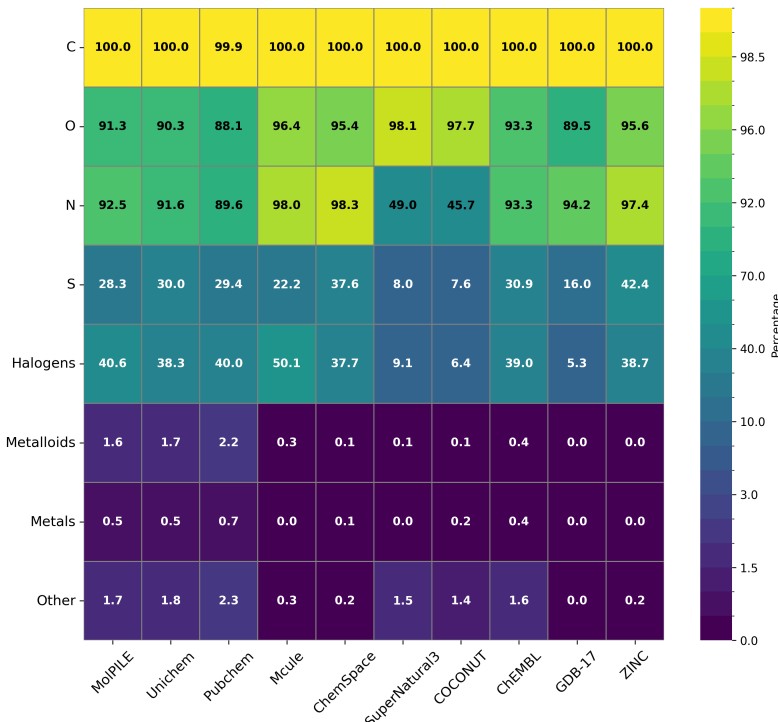

Figure 2: Percentage of molecules containing particular element or elements group.

## 4.2 ELEMENT GROUPS

The presence and frequency of different atomic elements, such as $C$, $O$, $N$, $S$, as well as halogens, metalloids, and metals, directly influence model capabilities. Pretraining on data lacking certain elements typically groups them as an "other" type, which can substantially reduce performance in relevant domains. For instance, Mol2vec (Jaeger et al., 2018), MAT (Maziarka et al., 2020), and R-MAT (Maziarka et al., 2024) were trained on a limited subset of elements, which strongly diminished their performance in ecotoxicology and agrochemistry (Adamczyk et al., 2025b). Conversely, pretraining on data that include metals and metalloids is crucial for ML models in areas like synthesis prediction (for example organometallic reactions in Suzuki coupling (Atz et al., 2024)) or oncological therapeutics (Peng et al., 2023). Thus, rich representation of element groups is an important indicator of both quality and diversity of molecular datasets.

Figure 2 illustrates the composition of element groups in MolPILE compared to other datasets. MolPILE exhibits a broad distribution, with a relatively high fraction of halogen-, metalloid-,

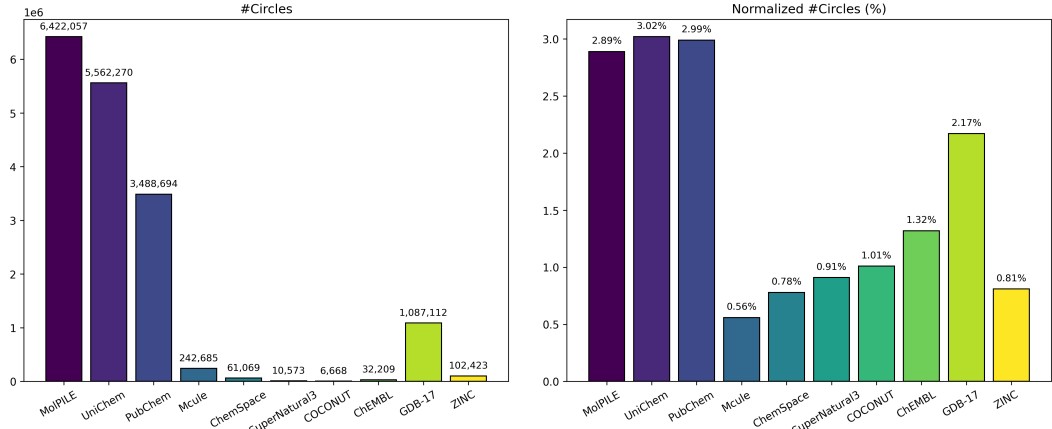

Figure 3: #Circles of each dataset and its normalized values, accounting for dataset size differences.

and metal-containing compounds. Its constituent datasets vary, but interestingly natural products in SuperNatural3 and COCONUT include far fewer nitrogen- and sulfur-containing compounds. ChEMBL, GDB-17, and ZINC are notably limited in elemental diversity. Specifically, ZINC and GDB-17 contain no metalloids or metals, ChEMBL includes very few, and GDB-17 heavily under-represents halogens. These patterns suggest that neural networks pretrained on ChEMBL or ZINC may underperform on chemical spaces beyond traditional medicinal chemistry, such as anti-tumor metallodrugs.

### 4.3 #CIRCLES STRUCTURAL DIVERSITY

Measuring chemical space coverage requires assessing how structurally representative the compounds in a dataset are. While one could enumerate an enormous space of nearly identical compounds by varying scaffolds minimally or substituting peripheral functional groups, such an approach would yield a dataset with intuitively low diversity. To address this, the #Circles metric was introduced in Xie et al. (2023) as a measure of diversity for generated molecular spaces, and it can likewise be applied to general compound datasets (Adamczyk et al., 2025a). Formally, it is defined as the maximum number of disjoint circles of radius $t$, with centers located at elements of the molecule set $\mathcal{S}$ (equivalent to the packing number in topology), as shown in Equation 1.

$$\#Circles(\mathcal{S}, d, t) = \max_{\mathcal{C} \subseteq \mathcal{S}} |\mathcal{C}| \quad \text{where} \quad d(x,y) \geq t \quad \forall x \neq y \in \mathcal{C} \tag{1}$$

To compare datasets of different sizes and assess their relative diversity, we also compute normalized #Circles by dividing the raw value by dataset size (Adamczyk et al., 2025a). Following the original formulation, we use binary ECFP fingerprints with Tanimoto distance and a threshold of $t = 0.75$. Although exact computation of #Circles is NP-hard (Mironov & Prokhorenkova, 2025), it can be efficiently approximated and parallelized (Xie et al., 2023).

Figure 3 presents the results for both raw and normalized #Circles. As expected, the raw metric scales with dataset size, with MolPILE achieving the largest value overall. Normalized #Circles, however, reveals differences in structural diversity independent of dataset size. Large-scale collections such as MolPILE, UniChem, and PubChem exhibit high diversity, each with values around 3%. In contrast, widely used ZINC shows remarkably low value of 0.81%. This is concerning for large-scale model training, as neural scaling laws require sufficiently diverse data to hold. Indeed, this observation may explain the performance saturation reported in ChemFM (Cai et al., 2024), where decoder-only transformers trained on ZINC plateaued starting at just 60M parameters. This phenomenon was not observed with PubChem when using the same number of compounds, likely due to its higher diversity. ChEMBL also shows limited diversity, with a normalized #Circles of 1.32%, only slightly better than ZINC.

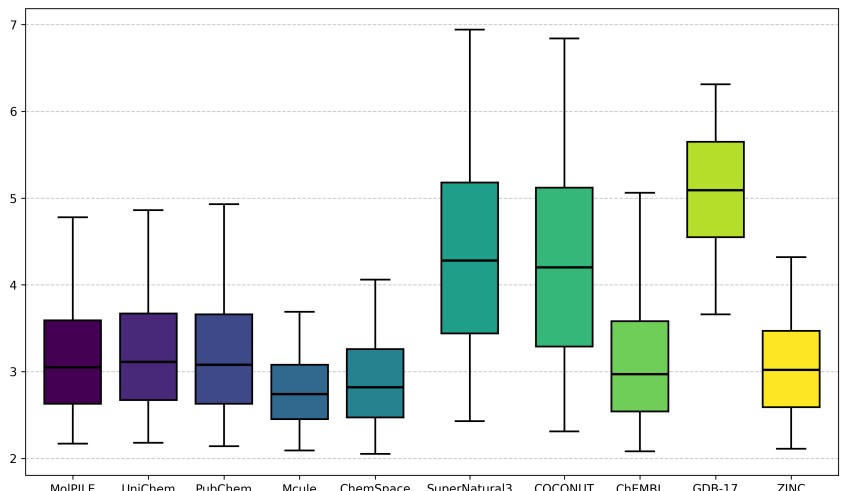

Figure 4: SAScore distribution in each dataset.

## 4.4 SYNTHESIZABILITY

The Synthesizability Score (SAScore) (Ertl & Schuffenhauer, 2009) summarizes the ease of synthesizing a given compound. It ranges from 1 to 10, with lower values indicating easier synthesis. Compared to alternative scoring systems, SAScore has been shown to yield more accurate and intuitive results (Skoraczyński et al., 2023; Neeser et al., 2024). Typically, compounds with SAScore above 5–6 are considered difficult to synthesize using standard techniques. The overall distribution of SAScore in a dataset therefore serves as a proxy for its practical quality, with realistic, experimentally synthesizable datasets expected to exhibit predominantly low values.

Figure 4 presents boxplots of SAScore values across datasets (see Appendix I for full statistics). GDB-17 exhibits markedly higher scores than other datasets, reflecting the general impracticality of synthesizing many enumerated compounds. This raises concerns for pretraining models, which may overfit to molecules that are challenging or unrealistic to access. In contrast, MolPILE shows a reasonable distribution with a median SAScore of 3.05, consistent with its construction from experimentally validated compounds. ChEMBL and ZINC also display sensible distributions. Slightly higher scores and heavy tails in SuperNatural3 and COCONUT are expected, as natural products are generally more difficult to synthesize using traditional methods (Ertl & Schuffenhauer, 2009).

## 4.5 FURTHER ANALYSES

We performed several additional analyses of MolPILE and alternative datasets. Due to space constraints, we summarize the main findings here and refer readers to the appendices for details.

Appendix J reports distributions of molecular descriptors used in our feasibility filter (e.g., molecular weight, logP, HBA, HBD). While most datasets follow expected patterns, ZINC stands out as highly homogeneous. It consists almost entirely of typical medicinal compounds consistent with textbook guidelines for orally bioavailable drugs, e.g., low weight, logP about 3, and very few HBD.

In Appendix K, we further analyze molecular filters, both physicochemical (e.g., Lipinski's rule of 5) and substructure-based (e.g., REOS). GDB-17 stands out by passing Lipinski's rule in 100% of cases, as well as GSK and Veber filters. Since molecular filters act as simple rule-based classifiers that restrict chemical space by predefined conditions, this outcome highlights the low diversity of GDB-17, reflecting its conservative enumerated design.

Appendix L reports diversity analyses in terms of Bemis–Murcko scaffolds (Bemis & Murcko, 1996), functional groups detected by Ertl's algorithm (Ertl, 2017), and salts. MolPILE exhibits broad chemical diversity, with over 3.6M unique scaffolds, 128k functional groups, and 1M salts. By comparison, GDB-17 and ZINC contain virtually no salts, while ChEMBL and ZINC include fewer than 6k unique functional groups, underscoring the limitations of existing datasets.

## 5 MODEL PRETRAINING

To practically evaluate the impact of MolPILE qualities, we retrain Mol2vec (Jaeger et al., 2018) and ChemBERTa (Ahmad et al., 2022) with it. We select those models, as they represent two significantly different models and approaches to preparing pretraining data. Further, many of the largest pretrained models for molecules have been SMILES-based, so we focus on this approach. Evaluating large-scale pretraining of graph neural networks on MolPILE is a direction for future work.

Originally, Mol2vec was trained on 20M molecules from ZINC15 and ChEMBL with stringent filtering: molecular weight $\in [12, 600]$, number of heavy atoms $\in [3, 50]$, logP $\in [-5, 7]$, removing counterions and solvents, as well as keeping only elements $[H, B, C, N, O, F, P, S, Cl, Br]$. ChemBERTa was originally trained on 77M molecules from PubChem, chosen randomly without filtering.

We retrained Mol2vec and ChemBERTa (MLM variant) on MolPILE, closely following their original implementations. Inspired by Praski et al. (2025), we use those models as pretrained feature extractors, training a Random Forest model on top of their embeddings. We include a variety of benchmarks: MoleculeNet (Wu et al., 2018) and TDC (Huang et al., 2021) for molecular property prediction, WelQrate (Liu et al., 2024) for ligand-based virtual screening, ToxBench dataset (Liu et al., 2025) for ligand-only binding prediction, as well as ApisTox dataset (Adamczyk et al., 2025a;b) representing ecotoxicology. In total, we use 48 individual datasets. We report the metric recommended for each benchmark by its authors: AUROC, MAE, RMSE, or BEDROC (Truchon & Bayly, 2007). We report average gain in each benchmark, as well as the number of wins of the retrained model. See Appendix M for details on datasets, training procedures, and all results on individual datasets.

Results are summarized in Table 2, showing that retraining on MolPILE consistently improves performance across the board. Even the relatively simple Mol2vec model, which lacks many parameters to fully exploit an 11 times larger dataset, shows clear gains. This emphasizes the importance of dataset variety and reasonable filtering for generalization. ChemBERTa likewise achieves substantial improvements, particularly on TDC datasets. The only exception is its weaker performance on ApisTox, likely due to the dataset's atypical and extremely limited vocabulary of just 600 tokens. To benefit from larger and more diverse datasets, ChemBERTa would need to scale up the tokenizer. In contrast, Mol2vec, which does not constrain vocabulary size, achieves strong gains on this agrochemical dataset. Notably, both models perform much better on MoleculeNet, TDC, and WelQrate, indicating that greater data quantity and diversity also benefit ADMET and virtual screening tasks.

Table 2: Results of original and retrained Mol2vec and ChemBERTa.

| Model | Measure | MoleculeNet | | TDC | | WelQrate | ToxBench | ApisTox |
|-------|---------|-------------|-----|-----|-----|----------|----------|---------|
| | | AUROC ↑ | MAE ↓ | AUROC ↑ | MAE ↓ | BEDROC ↑ | RMSE ↓ | AUROC ↑ |
| Mol2vec | Avg. gain | +1.79 | -0.203 | +0.58 | -0.496 | +1.52 | -0.139 | +2.23 |
| | # of wins | 3/8 | 3/3 | 15/18 | 4/8 | 7/9 | 1/1 | 1/1 |
| ChemBERTa | Avg. gain | +1.09 | -0.200 | +1.26 | -1.417 | +1.07 | -0.017 | -2.81 |
| | # of wins | 6/8 | 3/3 | 13/18 | 5/8 | 5/9 | 1/1 | 0/1 |

## 6 CONCLUSIONS

In this work, we introduced MolPILE, a large-scale dataset of nearly 222 million small molecules, containing experimentally validated and synthesizable compounds. Through a multi-step workflow, we ensured high-quality, deduplicated, and standardized molecular structures. Our analyses demonstrate that MolPILE offers high diversity, broad chemical space coverage, and superior structural quality compared to widely used datasets such as ChEMBL, GDB-17, and ZINC. We further show that these qualities translate into improved generalization of ML models, with Mol2vec and ChemBERTa pretrained on MolPILE achieving superior performance. MolPILE is released in SMILES format, along with curated, diversity-focused subsets, to facilitate accessible benchmarking. We believe MolPILE will support the development of higher-quality and broadly applicable molecular representation learning models, and serve as a standardized resource similar to ImageNet and PILE.

# 7 REPRODUCIBILITY STATEMENT

All code for reproducing the results is provided in the supplementary material. Scripts for creating MolPILE work end-to-end, downloading and processing the data. Similarly, we include scripts for working with ChEMBL, GDB-17, and ZINC. Code also contains scripts to retrain Mol2vec and ChemBERTa, and evaluate them on benchmarks. All dependencies are managed with `uv` manager, and we also include `uv.lock` file with all direct and transitive dependency versions. Instructions for setup and running all elements are included in the README. All analyses presented as plots or summaries in the main body have been expanded and detailed in appendices, including full results tables. Additional descriptions of datasets, models, algorithms (including diverse subset picking for MolPILE subsets), and implementation and hardware used, are included in appendices.

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

# A PRETRAINING DATASETS OF CURRENT MODELS

Table 3: Overview of pretraining datasets of molecular representation learning models.

| Model group | Model | Data source(s) | Dataset size | Data filtering |
|---|---|---|---|---|
| Graph neural networks (GNNs) | ContextPred (Hu et al., 2020) | ZINC / ChEMBL (two stages) | 2M / 456k (two stages) | - |
| | GEM (Fang et al., 2022) | ZINC | 20M | - |
| | GraphMVP (Liu et al., 2022) | GEOM | 50k | - |
| | MolR (Wang et al., 2022) | USPTO | 479k | - |
| Graph transformers | GROVER (Rong et al., 2020) | ZINC + ChEMBL | 11M | - |
| | MAT (Maziarka et al., 2020) | ZINC | 2M | - |
| | R-MAT (Maziarka et al., 2024) | ZINC + ChEMBL | 4M | Lipinski's rule of 5 (reduced data from 10M to 4M) |
| | Uni-Mol (Zhou et al., 2023) | ZINC + ChEMBL | 19M | - |
| SMILES transformers | CDDD (Winter et al., 2019) | ZINC + PubChem | 72M | organic molecules only molecular weight $\in [12, 600]$ heavy atoms $\geq 3$ logP $\in [-7, 5]$ removed steochemistry salts stripped to largest fragments |
| | ChemBERTa (Ahmad et al., 2022) | PubChem | 5M / 10M / 77M (three variants) | - |
| | ChemFM (Cai et al., 2024) | UniChem | 178M | - |
| | Chemformer (Irwin et al., 2022) | ZINC | 100M | reactivity "reactive" purchasability "annotated" molecular weight $\leq 500$ logP $\leq 5$ |
| | MolBERT (Fabian et al., 2020) | GuacaMol (ChEMBL subset) | 1.6M | - |
| | MoLFormer (Ross et al., 2022) | ZINC + PubChem | ~1.11B | SMILES length $\leq 211$ |
| | SimSon (Lee et al., 2025) | PubChem | 1M | - |
| SELFIES transformers | ChemGPT (Frey et al., 2023) | PubChem | 10M | - |
| | SELFormer (Yüksel et al., 2023) | ChEMBL | 2M | - |
| Hybrid models | CLAMP (Seidl et al., 2023) | PubChem Assays | 2.1M | - |
| | COATI (Kaufman et al., 2024) | ChEMBL + GEOM-Drugs + TensorMol + Mcule + ZINC22 + Enamine building blocks | 159M | - |
| | Mol2vec (Jaeger et al., 2018) | ZINC + ChEMBL | 20M | RDKit parseable molecular weight $\in [12, 600]$ heavy atoms count $\in [3, 50]$ logP $\in [-5, 7]$ counterions removed solvents removed only elements $[H, B, C, N, O, F, P, S, Cl, Br]$ |

Table 3 summarizes 21 molecular representation learning models and their pretraining datasets, including data sources, sizes (number of molecules), and filtering rules (if any), as selected from the benchmarking study of Praski et al. (2025).

A large majority of models - 16 out of 21 - apply no filtering, thereby including many low-quality, atypical, duplicated, or erroneous molecules, which MolPILE explicitly removes. Our analyses show that these issues are particularly prevalent in PubChem, UniChem, and ZINC, at least one of which is used by 15 of the 21 models. By contrast, models such as Mol2vec, R-MAT, CDDD, and Chemformer apply strict filtering, but in doing so heavily restrict the explored chemical space. Notably, three models trained on the largest and most diverse datasets - MoLFormer, ChemFM, and COATI - apply no filtering at all, leaving structure viability and synthesizability unchecked. Furthermore, 9 models rely on datasets of fewer than 10 million molecules, which is insufficient to adequately represent chemical space and limits generalization capability.

Taken together, these observations reveal three major shortcomings in existing pretraining datasets: small size, low quality, and low diversity. Many models are trained on datasets too small to generalize broadly across chemistry; others rely on unfiltered datasets containing unrealistic, low-quality molecules; and a third group applies overly strict filtering, eliminating essential diversity. MolPILE addresses all three issues by providing a large-scale, high-quality, and diverse dataset, thereby meeting the key desiderata for pretraining in molecular representation learning.

## B UniChem and PubChem sources

Here, we provide an overview of the major databases contained within UniChem and PubChem. As listing all sources would be impractical, since both integrate data from hundreds of vendors, we highlight only the most significant ones. Because UniChem and PubChem are incorporated into MolPILE, their sources are included implicitly and do not need to be added separately to our pipeline. Although some sources appear in both databases, MolPILE's standardization and merging steps ensure proper handling of duplicates. Importantly, many UniChem sources are subject to strong filtering, and their inclusion within the MolPILE pipeline introduces numerous additional compounds.

Ten largest UniChem sources:

1. PubChem (subset)
2. Mcule (subset)
3. SureChEMBL
4. ZINC (subset)
5. eMolecules
6. Nikkaji
7. ChEMBL
8. BindingDB
9. EPA CompTox Dashboard
10. SwissLipids

Ten largest PubChem sources:

1. Aurora Fine Chemicals LLC
2. SureChEMBL
3. AKos Consulting & Solutions
4. PATENTSCOPE (WIPO)
5. Google Patents
6. IBM
7. ChemSpider (subset)
8. ZINC (subset)
9. DiscoveryGate
10. NextBio

Other sources of PubChem also include subsets from, e.g., Mcule, Enamine, ChemSpace, MolGenie, ChemDB, ChEMBL, ChemBank, Wikidata, eMolecules.

## C  MOLPILE PIPELINE IMPLEMENTATION AND HARDWARE

For implementing MolPILE, we use:

- uv for dependency management
- aria2 and rapidgzip (Knespel & Brunst, 2023) for files downloading and unpacking
- Joblib for multiprocessing
- Polars (Polars) and DuckDB (Raasveldt & Mühleisen, 2019) for data engineering and files interaction
- RDKit (Landrum et al., 2025) and scikit-fingerprints (Adamczyk & Ludynia, 2024) for chemoinformatics tasks
- Gensim (Řehůřek & Sojka, 2010) for implementing Mol2vec training
- HuggingFace Transformers (Wolf et al., 2020) and Datasets (Lhoest et al., 2021) for implementing ChemBERTa training
- scikit-learn (Pedregosa et al., 2011) for implementing and evaluating models and benchmarks

For calculating MolPILE and performing analyses, we used a server with 128-core CPU and 512 GB RAM. We have also validated that it works on an average-grade machine with 12-core CPU, 64 GB RAM, and additional swap memory. The entire end-to-end pipeline, including downloading source datasets, takes under 12 hours on a server, and under 48 hours on the average-grade PC.

## D  FEASIBILITY FILTER DESIGN

Design of *molecular feasibility filter* included three main components: analysis of distributions of descriptors in source datasets, consulting an expert chemist, and referencing the literature. We also compared those distributions to values of approved drugs from DrugBank. To recall from the main body, the final molecular feasibility filter criteria are:

- molecular fragments $\leq 3$
- length of InChI $< 2000$
- molecular weight $\leq 2500$
- number of atoms $\leq 150$
- HBA $\leq 20$
- HBD $\leq 15$
- logP in range $[-10, 25]$
- TPSA $\leq 500$
- number of rotatable bonds $\leq 60$

We primarily focused on analyzing the basic distribution statistics like min/max, quartiles, or very low/high percentiles, as this information summarizes the knowledge about tails of the distribution. Similar techniques are often used to design molecular filters (), but they are typically more specific, leaving only, e.g., lead-like molecules () or pesticides of specific type (). Here, we are interested in filtering out most extreme and unusual molecules based on that data, summarized in:

- molecular weight - Table 4
- number of atoms - Table 5
- hydrogen bond acceptors (HBA) - Table 6
- hydrogen bond donors (HBD) - Table 7

- water-octanol partition coefficient logarithm (logP) - Table 8
- topological polar surface area (TPSA) - Table 9
- number of rotatable bonds - Table 10

For readability, numbers are presented as integers where possible without loss of precision (HBA, HBD, number of rotatable bonds).

First, we limit the number of molecular fragments to at most 3 to keep only typical single-component molecules and salts. Complexes with over 3 fragments are extremely rare, atypical, and very unstable. In most cases, chemists wouldn't consider them "molecules" for the purpose of data processing, but rather complexes or mixtures.

We impose a few size limits to remove proteins, peptides and other polymers, large crystals, and metal complexes. This removes compounds too large to be considered "small molecules", which are of primary interest here. While peptide-based therapeutics are of interest, they require dedicated processing, algorithms, and datasets, incorporating more informatics specific for biologics and proteins (Eastman et al., 2023). Those conditions are length of InChI $< 2000$, molecular weight $\leq 2500$, and number of atoms $\leq 150$, consistent with literature (Lagorce et al., 2017a; Fourches et al., 2010). Note that those values are larger than 99th percentile of all source datasets, thus removing compounds only from the 1% of the most extreme outliers. Analyzing the DrugBank approved drugs, the 99th percentile of molecular weight and number of atoms are $1565.01$ and $103$, respectively, so we also cover those (Fourches et al., 2010). We also note that chemical LLMs based on SMILES or SELFIES typically impose the limit for sequence length of 512 tokens, due to quadratic complexity of attention mechanism (Ahmad et al., 2022; Irwin et al., 2022; Yüksel et al., 2023). Thus, removing those larger compounds ensures proper data quality and removing outliers, while having minimal impact on its size.

Another group of requirements is related to compound stability and synthesizability. We do not want to include molecules that are clearly too unstable and unobtainable under typical synthesis routes, which also negatively impacts their bioavailability, selectivity, cell permeability, and protein binding properties (Veber et al., 2002; Kenny, 2022; Coimbra et al., 2021). In extreme cases, molecules would not have any stable structure at all under viable conditions, making their behavior unpredictable and compound unusable in practice. Thus, we limit the hydrogen bond acceptors (HBA) to at most 20, HBD at most 15, and number of rotatable bonds $\leq 60$. Those requirements still allow very flexible molecules, covering 95th percentile of DrugBank allowed drugs (HBA 14, HBD 7, rotatable bonds 16), but we also allow larger values due to quite flexible RDKit definition of rotatable bonds.

We limit the TPSA to at most 500 to allow almost all natural products from SuperNatural3 and CO-CONUT to be included. This number also admits large industrial chemistry compounds and various molecules outside medicinal chemistry, not aiming to permeate cell barriers. The main pharmaceutical motivation for TPSA-based filtering is to ensure bioavailability and cell permeability, with even stricter requirements for more specialized drugs like those targeting central nervous system, which are recommended to be under 90 TPSA to permeate the blood-brain barrier (Kelder et al., 1999; Ertl et al., 2000). As MolPILE aims to be more general, covering not only medicinal chemistry, we allow quite large TPSA.

The logarithm of the water-octanol partition coefficient, logP, measures the compound lipophilicity, and is one of the key indicators of its potential ability to cross biological barriers. The typical range of values conducive to biological activity is between approximately 0 and 5, with high values indicating low solubility in an aqueous environment, and conversely low values suggesting limited ability to penetrate lipid membranes. It can usually be well-approximated computationally by a simple atomic contributions model of (Wildman & Crippen, 1999), and it is the only method available in RDKit. This method generally works well and for a wide variety of compounds. However, one should note that it has been designed primarily for small organic molecules, and its results may be subject to greater error outside this applicability domain, e.g. for larger biomolecules or specific classes of chemical compounds, such as polymers or inorganic compounds. logP values of 10 or higher are extremely rare, and compounds with such characteristics typically have no practical application as biologically active substances. Extreme values, such as logP values around 20, should generally be treated as computational artifacts rather than actual physicochemical parameters. Similar observations can be made for negative values, and those limitations also stem from physical

limitations of experimental methods (Soares et al., 2022). For those reasons, taking into account both the distribution of values of this descriptor and the reasonable limits observed in typical chemical measurements, a range of analysis from -10 to 25 was adopted.

Table 4: Molecular weight distributions statistics.

| Dataset | min | p1 | p5 | Q1 | mean | median | Q3 | p95 | p99 | max |
|---|---|---|---|---|---|---|---|---|---|---|
| UniChem | 1.0 | 154.0 | 205.3 | 289.4 | 407.9 | 356.4 | 435.0 | 725.9 | 1612.8 | 113821.4 |
| PubChem | 1.0 | 151.2 | 200.4 | 280.4 | 429.8 | 358.4 | 461.5 | 821.7 | 2092.0 | 113821.4 |
| Mcule | 9.0 | 188.6 | 250.3 | 345.4 | 395.5 | 393.5 | 448.5 | 528.6 | 595.7 | 6179.4 |
| ChemSpace | 13.0 | 189.2 | 247.3 | 314.2 | 360.9 | 352.4 | 401.5 | 488.4 | 580.5 | 8073.5 |
| SuperNatural3 | 1.0 | 150.2 | 222.4 | 345.4 | 512.0 | 439.6 | 600.8 | 1013.6 | 1444.8 | 5033.8 |
| COCONUT | 1.0 | 136.2 | 210.3 | 334.4 | 506.6 | 431.5 | 601.6 | 1009.1 | 1435.8 | 7860.7 |

Table 5: Number of atoms distributions statistics.

| Dataset | min | p1 | p5 | Q1 | mean | median | Q3 | p95 | p99 | max |
|---|---|---|---|---|---|---|---|---|---|---|
| UniChem | 1 | 10 | 14 | 20 | 28.2 | 25 | 30 | 51 | 113 | 999 |
| PubChem | 1 | 10 | 13 | 19 | 29.7 | 25 | 32 | 59 | 148 | 910 |
| Mcule | 1 | 13 | 17 | 24 | 28.1 | 28 | 32 | 38 | 43 | 419 |
| ChemSpace | 1 | 13 | 17 | 22 | 25.2 | 25 | 28 | 34 | 40 | 566 |
| SuperNatural3 | 1 | 10 | 16 | 25 | 36.4 | 31 | 43 | 72 | 100 | 352 |
| COCONUT | 1 | 10 | 15 | 24 | 36.0 | 31 | 43 | 71 | 100 | 551 |

Table 6: Hydrogen bond acceptors (HBA) distributions statistics.

| Dataset | min | p1 | p5 | Q1 | mean | median | Q3 | p95 | p99 | max |
|---|---|---|---|---|---|---|---|---|---|---|
| UniChem | 0 | 0 | 2 | 3 | 4.8 | 4 | 6 | 9 | 19 | 608 |
| PubChem | 0 | 0 | 1 | 3 | 5.0 | 4 | 6 | 10 | 23 | 729 |
| Mcule | 0 | 1 | 2 | 3 | 4.7 | 5 | 6 | 8 | 9 | 191 |
| ChemSpace | 0 | 1 | 2 | 4 | 4.8 | 5 | 6 | 8 | 9 | 121 |
| SuperNatural3 | 0 | 1 | 2 | 4 | 7.1 | 6 | 8 | 17 | 28 | 106 |
| COCONUT | 0 | 1 | 2 | 4 | 7.0 | 6 | 8 | 17 | 28 | 191 |

Table 7: Hydrogen bond donors (HBD) distributions statistics.

| Dataset | min | p1 | p5 | Q1 | mean | median | Q3 | p95 | p99 | max |
|---|---|---|---|---|---|---|---|---|---|---|
| UniChem | 0 | 0 | 0 | 1 | 1.5 | 1 | 2 | 4 | 8 | 444 |
| PubChem | 0 | 0 | 0 | 1 | 1.6 | 1 | 2 | 4 | 9 | 292 |
| Mcule | 0 | 0 | 0 | 1 | 1.1 | 1 | 1 | 3 | 4 | 116 |
| ChemSpace | 0 | 0 | 0 | 1 | 1.2 | 1 | 2 | 3 | 4 | 122 |
| SuperNatural3 | 0 | 0 | 0 | 1 | 3.0 | 2 | 4 | 10 | 17 | 80 |
| COCONUT | 0 | 0 | 0 | 1 | 2.9 | 2 | 4 | 10 | 17 | 119 |

Table 8: Water-octanol partition coefficient logarithm (logP) distributions statistics.

| Dataset | min | p1 | p5 | Q1 | mean | median | Q3 | p95 | p99 | max |
|---|---|---|---|---|---|---|---|---|---|---|
| UniChem | -2048.5 | -1.3 | 0.5 | 2.2 | 3.9 | 3.3 | 4.6 | 8.4 | 20.7 | 508.3 |
| PubChem | -2234.1 | -1.5 | 0.5 | 2.2 | 4.3 | 3.5 | 4.9 | 10.3 | 24.7 | 709.2 |
| Mcule | -83.7 | 0.3 | 1.5 | 3.1 | 4.1 | 4.1 | 5.1 | 6.4 | 7.5 | 61.4 |
| ChemSpace | -49.1 | -0.1 | 0.9 | 2.2 | 3.2 | 3.2 | 4.1 | 5.6 | 6.9 | 47.3 |
| SuperNatural3 | -33.3 | -4.2 | -1.3 | 1.9 | 4.2 | 3.6 | 5.3 | 13.5 | 20.2 | 47.6 |
| COCONUT | -83.7 | -4.5 | -1.4 | 1.8 | 4.3 | 3.5 | 5.3 | 16.2 | 20.7 | 77.1 |

Table 9: Topological polar surface area (TPSA) distributions statistics.

| Dataset | min | p1 | p5 | Q1 | mean | median | Q3 | p95 | p99 | max |
|---|---|---|---|---|---|---|---|---|---|---|
| UniChem | 0.0 | 0.0 | 20.2 | 49.3 | 78.4 | 69.4 | 91.7 | 151.9 | 323.7 | 16013.0 |
| PubChem | 0.0 | 0.0 | 16.4 | 45.7 | 80.1 | 68.1 | 93.5 | 165.8 | 388.8 | 11400.0 |
| Mcule | 0.0 | 18.8 | 33.1 | 55.6 | 73.7 | 71.5 | 89.4 | 118.8 | 145.6 | 3038.9 |
| ChemSpace | 0.0 | 21.1 | 35.0 | 56.5 | 74.4 | 72.4 | 89.4 | 118.5 | 146.1 | 3537.2 |
| SuperNatural3 | 0.0 | 9.2 | 26.3 | 63.2 | 117.7 | 89.9 | 140.6 | 296.1 | 500.1 | 2231.2 |
| COCONUT | 0.0 | 4.4 | 26.3 | 61.8 | 116.1 | 86.2 | 136.7 | 297.6 | 503.7 | 3548.4 |

Table 10: Number of rotatable bonds distributions statistics.

| Dataset | min | p1 | p5 | Q1 | mean | median | Q3 | p95 | p99 | max |
|---|---|---|---|---|---|---|---|---|---|---|
| UniChem | 0 | 0 | 1 | 3 | 6.4 | 5 | 7 | 13 | 43 | 784 |
| PubChem | 0 | 0 | 1 | 3 | 6.6 | 5 | 7 | 15 | 46 | 648 |
| Mcule | 0 | 1 | 2 | 4 | 5.6 | 5 | 7 | 10 | 13 | 228 |
| ChemSpace | 0 | 1 | 2 | 3 | 4.8 | 5 | 6 | 8 | 10 | 200 |
| SuperNatural3 | 0 | 0 | 1 | 3 | 9.0 | 5 | 9 | 41 | 55 | 148 |
| COCONUT | 0 | 0 | 0 | 3 | 9.4 | 5 | 9 | 46 | 57 | 224 |

# E    MOLPILE SOURCE DATASETS CONTRIBUTIONS

Table 11: Cross-dataset counts after filtering.

| Dataset | UniChem | PubChem | Mcule | ChemSpace | SuperNatural3 | COCONUT |
|---|---|---|---|---|---|---|
| UniChem | 0 | 72636030 | 172822096 | 176788176 | 182865354 | 183310907 |
| PubChem | 6068050 | 0 | 106906633 | 110323432 | 116332956 | 116765767 |
| Mcule | 32361592 | 33014109 | 0 | 40134274 | 43262835 | 43324997 |
| ChemSpace | 726029 | 829265 | 4532631 | 0 | 7737809 | 7773204 |
| SuperNatural3 | 100482 | 136064 | 958467 | 1035084 | 0 | 590154 |
| COCONUT | 48647 | 71487 | 523241 | 573091 | 92766 | 0 |

In Table 11, we present a cross-dataset new molecule counts. In $i$-th row and $j$-th column, we show the number of molecules present in dataset $i$, but not in $j$-th. Each pair of datasets contains a non-overlapping subset, meaning that each database brings further new molecules to MolPILE.

# F    LICENSING

MolPILE is a collection of processed datasets, that we redistribute. It is shared as a single Parquet file with columns *source* and *ID*, where source is e.g. "PubChem" or "UniChem", and ID is the original identifier in the given database, e.g. PubChem CID. Each source has its own separate license, which we list below. As its entirety, MolPILE does not have a single license, as it is a collection, not a single dataset. Users interested in that can easily filter the dataset by source. In case of PubChem and UniChem, users may also want to check the individual licenses of their sources. Users using those sources are also asked to cite the appropriate publications. We do not make any claims about licensing of models trained on MolPILE, nor put any additional limitations.

Licensing of source datasets:

- PubChem (Kim et al., 2015) - CC0 (public domain)
- UniChem (Chambers et al., 2013) - CC-BY-4.0
- Mcule (Kiss et al., 2012) - CC-BY-NC-4.0
- ChemSpace (ChemSpace) - CC-BY-NC-4.0
- SuperNatural3 (Gallo et al., 2022) - not specified, only "freely available"
- COCONUT (Sorokina et al., 2021) - CC0 (public domain)

Table 12: Tanimoto distance distribution for 100k subset.

| Statistic | Random subset | Diverse subset | Difference |
|---|---|---|---|
| p10 | 0.881 | 0.915 | 0.034 |
| p25 | 0.901 | 0.939 | 0.038 |
| mean | 0.920 | 0.962 | 0.042 |
| median | 0.922 | 0.967 | 0.045 |
| p75 | 0.940 | 0.995 | 0.055 |
| p90 | 0.957 | 1.000 | 0.043 |
| Wasserstein distance | 0.042 | | - |
| K-S statistic | 0.507 | | - |

## G  DIVERSE SUBSETS PICKING

The complete MolPILE dataset contains 222 million molecules, which may be too large for initial experimentation or for training computationally expensive models, such as those based on contrastive learning. Selecting a random subset would sample proportionally to the local density of chemical space, thereby potentially biasing the model toward more easily synthesizable compounds, primarily from synthetic medicinal chemistry. To address this, and inspired by diversity selection methods used in virtual screening, we designed a procedure to construct diverse subsets. Specifically, we generated subsets of 100K, 1M, 5M, and 10M molecules.

The algorithm is based on prototype clustering using maximum diversity picking. Formally, the goal is to select a subset of $M$ compounds from a dataset of size $N$. First, $K$ compounds are chosen as cluster centers using the maximum diversity picking (Ashton et al., 2002) algorithm with minimum distance $t$. This ensures that the subset achieves maximal diversity, distributing the centers uniformly across the chemical space, so that the sum of their pairwise distances is maximized. Next, each of the remaining $N - K$ molecules is assigned to the nearest cluster center based on the Tanimoto distance. From each cluster, we select its center and randomly choose $\lfloor \frac{M-K}{K} \rfloor - 1$ additional molecules. If each cluster contains at least $\frac{M-K}{K}$ compounds, the algorithm terminates. In cases of highly uneven data density, where some clusters are much smaller than others, the subset may not reach size $M$. In such cases, the remaining molecules are randomly sampled from the unselected pool until the subset reaches the desired size.

As maximum diversity picking problem is NP-hard (Ghosh, 1996), we use the MaxMin approximation available in RDKit (Sayle). A distance threshold of $t = 0.9$ is applied, ensuring a high degree of separation of cluster centers. Under these settings, MolPILE contains $K = 7457$ cluster centers.

To verify the impact of this procedure, we compare it to randomly selected subsets of the same size. To quantify diversity, we compute the distribution of pairwise Tanimoto distances among molecules within each subset. For computational efficiency, we sample 100 million pairs to approximate the distribution, which should be sufficient; for example, authors of Vogt & Bajorath (2020) demonstrated that typically just 1 million pairs is sufficient to approximate well. We summarize the distributions using percentiles, mean, and median, and additionally compute the Wasserstein distance and the Kolmogorov-Smirnov (K-S) statistic between the two distributions. Note that larger values indicate that molecules are more distant. Subset statistics are reported in Table 12 for 100K, Table 13 for 1M, Table 14 for 5M, and Table 15 for 10M molecules.

The diverse subsets exhibit higher inter-molecule Tanimoto distances across the entire distribution. This is further supported by the relatively high Wasserstein distance (bounded by 1, like the Tanimoto distance) and the very high K-S statistic. Together, these results indicate that the diverse subsets closely approximate the highly diverse nature of MolPILE and should therefore be preferred over random subsets. Using standardized subsets further increases reproducibility of results for models trained on MolPILE.

Table 13: Tanimoto distance distribution for 1M subset.

| Statistic | Random subset | Diverse subset | Difference |
|---|---|---|---|
| p10 | 0.881 | 0.904 | 0.023 |
| p25 | 0.901 | 0.926 | 0.025 |
| mean | 0.920 | 0.948 | 0.028 |
| median | 0.921 | 0.950 | 0.029 |
| p75 | 0.940 | 0.974 | 0.034 |
| p90 | 0.957 | 1.000 | 0.043 |
| Wasserstein distance | 0.029 | | - |
| K-S statistic | 0.359 | | - |

Table 14: Tanimoto distance distribution for 5M subset.

| Statistic | Random subset | Diverse subset | Difference |
|---|---|---|---|
| p10 | 0.881 | 0.896 | 0.015 |
| p25 | 0.901 | 0.917 | 0.016 |
| mean | 0.920 | 0.938 | 0.018 |
| median | 0.921 | 0.939 | 0.018 |
| p75 | 0.940 | 0.962 | 0.022 |
| p90 | 0.957 | 0.981 | 0.024 |
| Wasserstein distance | 0.019 | | - |
| K-S statistic | 0.241 | | - |

Table 15: Tanimoto distance distribution for 10M subset.

| Statistic | Random subset | Diverse subset | Difference |
|---|---|---|---|
| p10 | 0.881 | 0.893 | 0.012 |
| p25 | 0.901 | 0.913 | 0.012 |
| mean | 0.920 | 0.934 | 0.014 |
| median | 0.921 | 0.935 | 0.014 |
| p75 | 0.940 | 0.957 | 0.017 |
| p90 | 0.957 | 0.975 | 0.018 |
| Wasserstein distance | 0.014 | | - |
| K-S statistic | 0.189 | | - |

## H    PIPELINE MOLECULE FILTERING DETAILS

In Table 16, we present exact numbers for all datasets and pipeline steps, a detailed version of Table 1 from the main body.

Table 16: Statistics of datasets: number of molecules removed at each step and the final dataset size.

| Dataset | Initial count | Preprocessing | Standardization | Filtering | Final dataset |
|---|---|---|---|---|---|
| UniChem | 189058653 | 0 | -458698 | -4677263 | 183922692 |
| PubChem | 121440975 | -466660 | -107100 | -4233834 | 116633381 |
| Mcule | 43580777 | -104840 | -157 | -13592 | 43462188 |
| ChemSpace | 7831419 | -78 | -107 | -4345 | 7826889 |
| SuperNatural3 | 1205199 | -2913 | -331 | -44135 | 1157820 |
| COCONUT | 695133 | -8584 | -154 | -25963 | 660432 |
| ChEMBL | 2.4M | -8 | -911 | -41393 | 2.4M |
| GDB-17 | 50M | -4267 | -5 | 0 | 50M |
| ZINC | 13.7M | -944883 | -8743 | -3026 | 12.7M |
| **MolPILE** | | | | | 221950487 |

## I    FULL SASCORE TABLE

Table I contains distribution information for SASCore values: minimum, maximum, mean, median, Q1 and Q3, and percentiles 1, 5, 95 and 99. Those results are for filtered datasets

Table 17: Synthesizability Score (SASCore) distributions

| Dataset | min | p1 | p5 | Q1 | mean | median | Q3 | p95 | p99 | max |
|---|---|---|---|---|---|---|---|---|---|---|
| UniChem | 1 | 1.91 | 2.18 | 2.67 | 3.26 | 3.11 | 3.67 | 4.86 | 6.08 | 10 |
| PubChem | 1 | 1.87 | 2.14 | 2.63 | 3.25 | 3.08 | 3.66 | 4.93 | 6.22 | 10 |
| Mcule | 1 | 1.86 | 2.09 | 2.45 | 2.80 | 2.74 | 3.08 | 3.69 | 4.33 | 8.92 |
| ChemSpace | 1 | 1.80 | 2.05 | 2.47 | 2.91 | 2.82 | 3.26 | 4.06 | 4.85 | 9.47 |
| SuperNatural3 | 1 | 1.98 | 2.43 | 3.44 | 4.42 | 4.28 | 5.18 | 6.94 | 7.97 | 9.53 |
| COCONUT | 1 | 1.88 | 2.31 | 3.29 | 4.31 | 4.20 | 5.12 | 6.84 | 7.92 | 9.53 |
| **MolPILE** | 1 | 1.92 | 2.17 | 2.63 | 3.20 | 3.05 | 3.59 | 4.78 | 6 | 10 |
| ChEMBL | 1 | 1.81 | 2.08 | 2.54 | 3.18 | 2.97 | 3.58 | 5.06 | 6.50 | 9.01 |
| GDB-17 | 1 | 3.14 | 3.66 | 4.55 | 5.07 | 5.09 | 5.65 | 6.31 | 6.73 | 8.19 |
| ZINC | 1 | 1.84 | 2.11 | 2.59 | 3.08 | 3.02 | 3.47 | 4.32 | 5.10 | 7.99 |

## J    MOLECULAR DESCRIPTORS ANALYSIS

In this section, we analyze the distributions of molecular descriptors in MolPILE and other datasets used for model pretraining: ChEMBL, GDB-17, and ZINC. Distribution statistics are provided in Tables 18, 19, 20, 21, 22, 23 and 24. Values for ChEMBL, GDB-17 and ZINC are provided without the feasibility filter to avoid skewing the results.

First, ChEMBL clearly contains some outliers and unreasonable molecules, as evidences by maximum value in each descriptor, e.g. maximum molecular weight about 12.5 thousand daltons or 360 rotatable bonds. GDB-17 shows very low numbers in all regards, which is a consequence of its combinatorial construction and using at most 17 atoms. ZINC shows very low diversity and contains a very conservative set of typical medicinal compounds. Its highest molecular weight is under 1000 daltons, and the largest number of atoms is 60, not covering e.g. many oncological drugs. Similarly, there are at most 20 HBA, 15 HBD, and 45 rotatable bonds, indicating a very conservative distribution in terms of structure flexibility. We note that this may also be due to our choice of a representative subset of ZINC with established 3D structures, but the general trend points to low diversity of that dataset in terms of physicochemical properties.

Table 18: Molecular weight distribution statistics.

| Dataset | min | p1 | p5 | Q1 | mean | median | Q3 | p95 | p99 | max |
|---|---|---|---|---|---|---|---|---|---|---|
| ChEMBL | 4.0 | 181.2 | 240.3 | 325.4 | 436.4 | 393.5 | 476.4 | 707.2 | 1457.8 | 12546.3 |
| GDB-17 | 30.1 | 192.3 | 207.2 | 224.3 | 236.3 | 237.3 | 242.3 | 262.3 | 320.2 | 703.8 |
| ZINC | 51.1 | 170.2 | 237.2 | 319.4 | 383.7 | 374.5 | 440.9 | 553.4 | 655.9 | 995.1 |
| MolPILE | 1.00 | 156.20 | 210.23 | 297.31 | 383.53 | 364.39 | 436.56 | 629.64 | 923.98 | 2500.00 |

Table 19: Number of atoms distribution statistics.

| Dataset | min | p1 | p5 | Q1 | mean | median | Q3 | p95 | p99 | max |
|---|---|---|---|---|---|---|---|---|---|---|
| ChEMBL | 1.0 | 12.0 | 17.0 | 23.0 | 30.6 | 28.0 | 34.0 | 50.0 | 102.0 | 780.0 |
| GDB-17 | 2.0 | 14.0 | 15.0 | 16.0 | 16.5 | 17.0 | 17.0 | 17.0 | 17.0 | 18.0 |
| ZINC | 4.0 | 12.0 | 16.0 | 22.0 | 26.6 | 26.0 | 31.0 | 38.0 | 44.0 | 60.0 |
| MolPILE | 1.00 | 10.00 | 14.00 | 20.00 | 26.71 | 25.00 | 31.00 | 44.00 | 66.00 | 150.00 |

Table 20: Hydrogen bond acceptors (HBA) distribution statistics.

| Dataset | min | p1 | p5 | Q1 | mean | median | Q3 | p95 | p99 | max |
|---|---|---|---|---|---|---|---|---|---|---|
| ChEMBL | 0.0 | 1.0 | 2.0 | 4.0 | 5.7 | 5.0 | 7.0 | 10.0 | 21.0 | 290.0 |
| GDB-17 | 0.0 | 1.0 | 2.0 | 3.0 | 4.3 | 4.0 | 5.0 | 6.0 | 7.0 | 12.0 |
| ZINC | 0.0 | 1.0 | 2.0 | 3.0 | 4.7 | 5.0 | 6.0 | 8.0 | 10.0 | 20.0 |
| MolPILE | 0.00 | 1.00 | 2.00 | 3.00 | 4.55 | 4.00 | 6.00 | 8.00 | 12.00 | 20.00 |

Table 21: Hydrogen bond donors (HBD) distribution statistics.

| Dataset | min | p1 | p5 | Q1 | mean | median | Q3 | p95 | p99 | max |
|---|---|---|---|---|---|---|---|---|---|---|
| ChEMBL | 0.0 | 0.0 | 0.0 | 1.0 | 2.1 | 1.0 | 2.0 | 5.0 | 19.0 | 121.0 |
| GDB-17 | 0.0 | 0.0 | 0.0 | 1.0 | 2.2 | 2.0 | 3.0 | 5.0 | 6.0 | 10.0 |
| ZINC | 0.0 | 0.0 | 0.0 | 0.0 | 1.1 | 1.0 | 2.0 | 3.0 | 4.0 | 15.0 |
| MolPILE | 0.00 | 0.00 | 0.00 | 1.00 | 1.37 | 1.00 | 2.00 | 3.00 | 6.00 | 15.00 |

Table 22: Water–octanol partition coefficient logarithm (logP) distribution statistics.

| Dataset | min | p1 | p5 | Q1 | mean | median | Q3 | p95 | p99 | max |
|---|---|---|---|---|---|---|---|---|---|---|
| ChEMBL | -247.5 | -1.4 | 0.7 | 2.6 | 3.8 | 3.7 | 4.9 | 7.0 | 9.9 | 180.5 |
| GDB-17 | -5.2 | -2.0 | -1.3 | -0.2 | 0.7 | 0.6 | 1.5 | 2.7 | 3.5 | 7.2 |
| ZINC | -10.0 | -0.9 | 0.4 | 2.2 | 3.4 | 3.5 | 4.6 | 6.4 | 7.8 | 17.7 |
| MolPILE | -10.00 | -0.83 | 0.66 | 2.36 | 3.73 | 3.48 | 4.68 | 7.31 | 14.02 | 25.00 |

Table 23: Topological polar surface area (TPSA) distribution statistics.

| Dataset | min | p1 | p5 | Q1 | mean | median | Q3 | p95 | p99 | max |
|---|---|---|---|---|---|---|---|---|---|---|
| ChEMBL | 0.0 | 12.9 | 30.0 | 56.9 | 97.0 | 78.3 | 104.8 | 188.7 | 552.0 | 4530.8 |
| GDB-17 | 0.0 | 15.0 | 29.5 | 55.4 | 74.5 | 73.8 | 93.2 | 121.4 | 141.0 | 216.4 |
| ZINC | 0.0 | 18.5 | 32.7 | 54.7 | 73.1 | 71.4 | 89.6 | 118.0 | 145.2 | 440.0 |
| MolPILE | 0.00 | 3.24 | 21.26 | 50.16 | 73.67 | 69.67 | 90.73 | 136.42 | 211.89 | 499.99 |

Table 24: Number of rotatable bonds distribution statistics.

| Dataset | min | p1 | p5 | Q1 | mean | median | Q3 | p95 | p99 | max |
|---------|-----|-----|-----|------|------|--------|------|-------|-------|--------|
| ChEMBL | 0.0 | 0.0 | 1.0 | 3.0 | 6.5 | 5.0 | 7.0 | 14.0 | 38.0 | 360.0 |
| GDB-17 | 0.0 | 0.0 | 0.0 | 1.0 | 2.3 | 2.0 | 4.0 | 6.0 | 8.0 | 13.0 |
| ZINC | 0.0 | 1.0 | 2.0 | 4.0 | 5.3 | 5.0 | 7.0 | 10.0 | 13.0 | 45.0 |
| MolPILE | 0.00 | 0.00 | 2.00 | 4.00 | 5.85 | 5.00 | 7.00 | 12.00 | 25.00 | 60.00 |

## K  MOLECULAR FILTERS ANALYSIS

Figure 5 presents the results of molecular filter analyses. Molecular filters are used in drug design for the initial selection of molecules based on their physicochemical and structural properties. They enable the rejection of compounds with unfavorable characteristics, such as low bioavailability, difficult synthesis or the presence of reactive fragments. This narrows down the chemical space to more promising candidates and increases the efficiency of further stages of drug design. However, they also limit the chemical space, which can decrease the innovation in terms of explored compound structures.

For readability, we restrict this comparison to MolPILE, ChEMBL, GDB-17, and ZINC across 16 filters (see below for their descriptions). Molecular filters can be broadly grouped into two categories: (i) drug-likeness filters, based on structural properties of typical medicinal chemistry compounds, and (ii) filters targeting other aspects such as toxicity, reactivity, or pesticide-likeness. Examining the fraction of molecules passing each filter provides complementary insight into both physicochemical properties and characteristic substructures of the datasets (Adamczyk et al., 2025b). Ideally, most molecules should pass standard filters, indicating consistency with well-established distributions. However, we also expect a subset to fall outside these boundaries, since advances in synthesis methods enable exploration of chemical space beyond conservative, rule-based definitions. In the case of physicochemical filters, this space corresponds directly to a descriptor-defined hypercube (e.g., molecular weight, logP), where strict adherence would imply limited diversity.

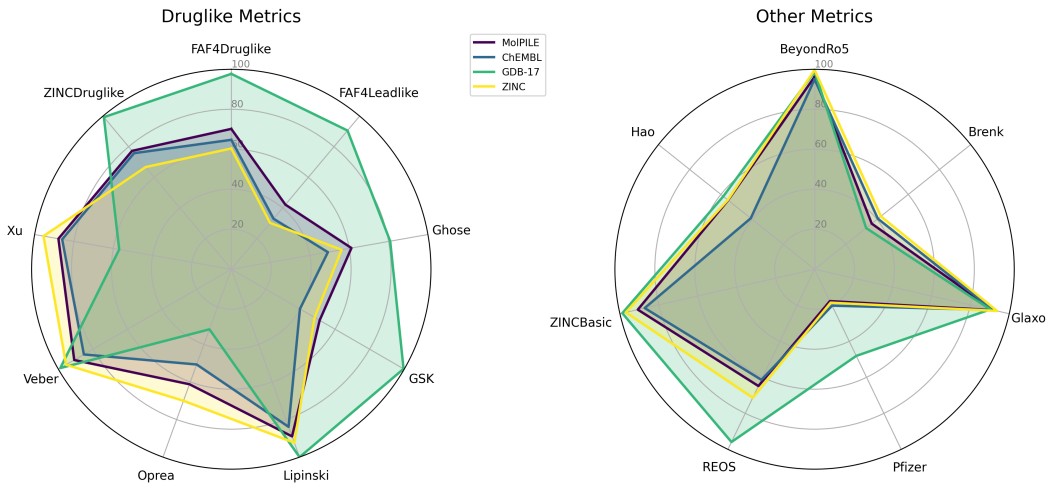

Figure 5: Percentage of molecules that fits to various molecular filters.

As expected, all datasets score highly on the Beyond Rule of 5 filter, which is designed to exclude only extreme, unreasonable molecules. This also supports the validity of our molecular feasibility filter. MolPILE shows balanced results overall, with the highest values on general filters such as Xu or Veber.

In contrast, GDB-17 fails to generalize beyond traditional chemical space, with near-complete compliance (about 100%) across multiple filters, including Lipinski, GSK, FAF4-druglike, ZINC-druglike, Veber, ZINC-basic, REOS, and Beyond Rule of 5. Other datasets display much lower pass

rates, particularly for more restrictive filters such as GSK, Ghose, FAF4-druglike, FAF4-leadlike, REOS, and ZINC-druglike.

Molecular filters used are:

1. **FAFDrugs4 drug-like filter** (Lagorce et al., 2017a) - designed to keep only drug-like molecules, based on literature describing physico-chemical properties of drugs and their statistical analysis. Selected so that up to 90% of the 916 FDA-approved oral drugs fulfill the rules of this filter. Molecules must fulfill conditions: molecular weight in range $[100, 600]$, logP in range $[-3, 6]$, HBA $\leq 12$, HBD $\leq 7$, TPSA $\leq 180$, number of rotatable bonds $\leq 11$, number of rigid bonds $\leq 30$, number of rings $\leq 6$, max ring size $\leq 18$, number of carbons in range $[3, 35]$, number of heteroatoms in range $[1, 15]$, non-carbons to carbons ratio in range $[0.1, 1.1]$, number of charged functional groups $\leq 4$, total formal charge in range $[-4, 4]$.

2. **ZINC drug-like** (Irwin & Shoichet, 2005) - designed to keep only drug-like molecules. Based only on physico-chemical properties, since SMARTS for additional rules are not publicly available. Molecules must fulfill conditions: molecular weight in range $[60, 600]$, logP in range $[-4, 6]$, HBA $\leq 11$, HBD $\leq 6$, TPSA $\leq 150$, number of rotatable bonds $\leq 12$, number of rigid bonds $\leq 50$, number of rings $\leq 7$, max ring size $\leq 12$, number of carbons $\geq 3$, non-carbons to carbons ratio $\leq 2.0$, number of charged functional groups $\leq 4$, total formal charge in range $[-4, 4]$.

3. **Rule of Xu** (Xu & Stevenson, 2000) - another filter designed to identify drug-like molecules. Molecules must fulfill conditions: HBD $\leq 5$, HBA $\leq 10$, number of rotatable bonds in range $[2, 35]$, number of rings in range $[1, 7]$, number of heavy atoms in range $[10, 50]$.

4. **Rule of Veber** (Veber et al., 2002) - designed to identify molecules likely to exhibit good oral bioavailability. Molecules must fulfill conditions: number of rotatable bonds $\leq 10$, TPSA $\leq 140$.

5. **Oprea filter** (Oprea, 2000) - filter for drug likeness, designed by comparing drug and non-drug compounds across multiple datasets. Molecules must fulfill conditions: HBD $\leq 2$, HBA in range $[2, 9]$, number of rotatable bonds in range $[2, 8]$, number of rings in range $[1, 4]$.

6. **Lipinski's Rule of 5 (RO5)** (Lipinski et al., 2001) - evaluates the drug-likeness of a molecule as an orally active drug, assuming that they are small and lipophilic. Also known as Pfizer's rule of 5. Molecules are allowed to violate at most one of the rules. Conditions: molecular weight $\leq 500$, HBA $\leq 10$, HBD $\leq 5$, logP $\leq 5$. Hydrogen bond acceptors (HBA) and donors (HBD) use a simplified definition, taking into consideration only oxygen and nitrogen bonds with hydrogen (OH, NH).

7. **GSK rule (4/400)** (Gleeson, 2008) - interpretable ADMET rule of thumb for drug-likeness. Molecules must fulfill conditions: molecular weight $\leq 400$, logP $\leq 4$ .

8. **Ghose filter** (Ghose et al., 1998) - used for searching for drug-like compounds. Molecules must fulfill conditions: molecular weight in range $[160, 400]$, logP in range $[-0.4, 5.6]$, number of atoms in range $[20, 70]$, molar refractivity in range $[40, 130]$ .

9. **FAFDrugs4 lead-like filter** (Lagorce et al., 2017b) - based on literature describing physico-chemical properties of lead drugs, designed to find lead-like candidates, i.e. starting point molecules that can be further optimized. They should be relatively small, with low logP, and can be "decorated" further to increase affinity and/or selectivity, without becoming very ADMET-unfriendly. Basically a more restrictive variant of FAFDrugs4 drug-like filter. Molecules must fulfill conditions: molecular weight in range $[150, 400]$, logP in range $[-3, 4]$, HBA $\leq 7$, HBD $\leq 4$, TPSA $\leq 160$, number of rotatable bonds $\leq 9$, number of rigid bonds $\leq 30$, number of rings $\leq 4$, max ring size $\leq 18$, number of carbons in range $[3, 35]$, number of heteroatoms in range $[1, 15]$, non-carbons to carbons ratio in range $[0.1, 1.1]$, number of charged functional groups $\leq 4$, total formal charge in range $[-4, 4]$, number of stereocenters $\leq 2$.

10. **Beyond rule of 5** (Doak et al., 2015) - designed to cover cover novel orally bioavailable drugs that do not fulfill the original conditions of Lipinski's rule of 5. Allows less typical

molecules, particularly suitable for difficult targets, allowing greater flexibility. Molecules must fulfill conditions: molecular weight $\leq 1000$, logP in range $[-2, 10]$, HBA $\leq 15$, HBD $\leq 6$, TPSA $\leq 250$, number of rotatable bonds $\leq 20$ .

11. **Hao rule for pesticides** (Hao et al., 2011) - designed to describe physicochemical properties of pesticides, for use in general pesticide design. Molecules must fulfill conditions: molecular weight $\leq 435$, logP $\leq 6$, HBD $\leq 2$, HBA $\leq 6$, number of rotatable bonds $\leq 9$, number of aromatic bonds $\leq 17$ .

12. **ZINC basic filter** (docking.org team) - designed to keep only drug-like molecules, removing molecules with unwanted functional groups. Used by docking.org for ZINC database as basic set of filters, applied to all vendor catalogs. Substructural filter, with rules available at docking.org team.

13. **REOS filter** (Walters & Namchuk, 2003) - Rapid Elimination Of Swill (REOS) filter is designed to remove molecules with undesirable properties for general drug discovery. Molecules must fulfill conditions: molecular weight in range $[200, 500]$, logP in range $[-5, 5]$, HBA $\leq 10$, HBD $\leq 5$, charge in range $[-2, 2]$, number of rotatable bonds $\leq 8$, number of heavy atoms in range $[15, 50]$.

14. **Pfizer 3/75 rule** (Hughes et al., 2008; Price et al., 2009) - based on observation that compounds exhibiting low partition coefficient (logP) and high topological polar surface area (TPSA) are roughly 2.5 times more likely to be free of toxicity issues in the tested conditions. Molecules must fulfill conditions: logP $\leq 3$, TPSA $\geq 75$.

15. **Glaxo filter** (Hann et al., 1999) - designed to filter out molecules with reactive functional groups, unsuitable leads (i.e., compounds which would not be initially followed up), and unsuitable natural products (i.e., derivatives of natural product compounds known to interfere with common assay procedures). Rule definitions are available in the supplementary material of the original publication (Hann et al., 1999).

16. **Brenk filter** (Brenk et al., 2008) - designed to filter out molecules containing substructures with undesirable pharmacokinetics or toxicity, e.g., sulfates, phosphates, nitro groups. Resulting set should be reasonable lead-like molecules for optimization campaigns and HTS. Rule definitions are available in the supplementary material of the original publication (Brenk et al., 2008).

## L  FUNCTIONAL GROUPS, SCAFFOLDS, AND SALTS ANALYSES

Table 25: Number of unique scaffolds, functional groups, and salts in each dataset.

| Dataset | Scaffolds | Functional groups | Salts |
|---|---|---|---|
| UniChem | 3,213,417 | 122,050 | 953,082 |
| PubChem | 3,046,726 | 79,103 | 784,495 |
| Mcule | 310,894 | 4,317 | 28,888 |
| ChemSpace | 259,597 | 3,140 | 9,841 |
| SuperNatural3 | 47,484 | 2,498 | 233 |
| COCONUT | 45,268 | 1,595 | 2,002 |
| **MolPILE** | 3,620,809 | 128,347 | 1,089,501 |
| ChEMBL | 172,606 | 3,828 | 22,474 |
| GDB-17 | 70,377 | 24,274 | 0 |
| ZINC | 191,514 | 5,813 | 9 |

Here, we present an analysis of Bemis-Murcko scaffolds (Bemis & Murcko, 1996), functional groups computed with Ertl's algorithm (Ertl, 2017), and salts. Those are structural measures of diversity, focusing on different aspects of molecular graph shape. Results are summarized in Table 25.

Number of unique Bemis-Murcko scaffolds is a commonly used diversity measure. Here, we count so-called "generic scaffolds", as described in the original paper (Bemis & Murcko, 1996), consisting of connected rings backbone, and replacing all atoms by carbons. This ensures that we focus solely

on the true core of the molecule. MolPILE has the largest number of scaffolds, over 1 million, showing its very high diversity in this regard. The absolute number here is important, as this means that models trained on MolPILE will be exposed to a wide collection of molecular shapes.

For functional groups analysis, we used Ertl's algorithm, which extracts them algorithmically. There is no one commonly used list of functional groups, and their definitions vary between chemists, so relying on an algorithmic approach is more objective in this regard. We select only functional groups between 2 and 20 atoms, and only those appearing in at least 10 molecules, in order to remove computational artifacts, similar to analysis of ChEMBL in the original paper (Ertl, 2017). Again, MolPILE contains the largest number of functional groups, which means that models pretrained on it will be exposed to various substituents. This is particularly relevant to SMILES-based transformers, as this will also impact the tokenization.

Lastly, we checked the number of salts, i.e., molecules with at least two charged disconnected components (in terms of covalent bonds). This is very important for SMILES-based transformers, as without this they won't even recognize the fact of a molecule having multiple fragments. The SMILES format uses the dot symbol "." to mark disconnected components, and it must be a part of the tokenizer and have a reasonable learned embedding in order to be useful. Interestingly, GDB-17 does not contain any salts at all, and ZINC only trace number of 9. This means that models trained on those datasets won't work on salts at all, while this is highly relevant to agrochemistry (Young et al., 2008; Adamczyk et al., 2025a). Such models may even error out on such SMILES strings, depending on the implementation. MolPILE contains over 1 million salts, about $0.5\%$, which exposes all models to a reasonable number of them, and allows proper learning.

## M  EVALUATION DATASETS AND MODEL TRAINING

Here, we describe evaluation details and model training procedure in detail. We also provide results on individual datasets.

### M.1  DATASETS AND BENCHMARKS

To evaluate models retrained on MolPILE, we use a total of 52 datasets from 2 general benchmarks for drug design and ADMET (MoleculeNet, TDC), ligand-based virtual screening benchmark (WelQrate), and two datasets from two distinctly different areas: protein-ligand binding (ToxBench), and ecotoxicology (ApisTox).

**MoleculeNet** (Wu et al., 2018) is the most commonly used benchmark for evaluating ML models for molecular property prediction. It contains 16 datasets, 9 for classification and 7 for regression. However, in literature most commonly used are 8 classification datasets and 3 regression datasets, excluding PCBA (due to extreme size), PDBBind (due to relatively low quality as a protein-ligand binding dataset), and QM7, QM8 and QM9 (quantum chemistry, requiring dedicated models and 3D information). Thus, we use the following 11 datasets: BACE, BBBP, HIV (single task classification), ClinTox, MUV, SIDER, Tox21, ToxCast (multi-task classification), ESOL, FreeSolv, and Lipophilicity (regression). AUROC is used for all classification datasets, and MAE for all regression ones.

**Therapeutics Data Commons (TDC** (Huang et al., 2021) is a large collection of diverse tasks focusing on medicinal chemistry, covering a wide variety of ADMET tasks. We select all datasets between around 500 and 50000 molecules, to have reasonable data size for training and testing. This amounts to 19 datasets. See tables below for all dataset names. AUROC is used for all classification datasets, and MAE for all regression ones.

**WelQrate** (Liu et al., 2024) has been designed as a "gold standard" benchmark for ligand-based virtual screening. It consists of 9 large-scale datasets from 5 therapeutic target classes, which underwent expert curation and rigorous process of categorization, verification, filtering and cleaning. It evaluates the ability of models to not only predict bioactivity and protein-ligand, based only on the ligand structure, but also to rank them from most to least promising. This is measured by using the BEDROC metric (Boltzmann-enhanced discrimination of receiver operating characteristic) (Truchon & Bayly, 2007).

**ToxBench** (Liu et al., 2025) is a dataset based on AB-FEP calculations, covering 8770 ligand–ER$\alpha$ complexes with calculated binding free energies, some of which have been experimentally verified. In particular, it features carefully prepared train-test data splits, enabling a reliable assessment of models for protein-ligand binding. We use it with a ligand-only approach, similar to ChemProp in Liu et al. (2025), embedding the ligand and predicting the binding free energy for ER$\alpha$ protein. The recommended metric, as usual in protein-ligand binding, is RMSE.

**ApisTox** (Adamczyk et al., 2025a;b) is a dataset for predicting pesticides toxicity to honey bees. It uses binarized LD50, following US EPA recommendation, resulting in binary classification task. We use the time split, which uses the newest pesticides as the test set, approximating the realistic conditions of novel pesticide design. Following recommendations, we use AUROC as a metric.

## M.2 MODEL TRAINING

We retrain Mol2vec (Jaeger et al., 2018) following the original authors' implementation, based on Gensim, using CPU only. Embeddings have 300 dimensions, and are trained using Skip-gram approach for 5 epochs. For ChemBERTa (Ahmad et al., 2022), we use HuggingFace transformers and MLM modeling, exactly following the original authors' code. The only difference is that we ensure that the model sees the whole dataset at least once before early stopping, and we evaluate every 500 steps, rather than 50, as the original settings stopped training abnormally quickly.

We use the same classification head in all cases for fair comparison: Random Forest (RF) with 500 trees and entropy or MSE split function (depending on dataset). This choice was made since RF performs very well on average, does not require extensive hyperparameter tuning, and is commonly used in chemoinformatics. It also supports multitask datasets out-of-the-box in the scikit-learn implementation. We do not perform any hyperparameter tuning, and use validation data for training in datasets already pre-split into training-validation-testing subsets (e.g. MoleculeNet, TDC).

## M.3 DETAILED RESULTS

Tables below contain detailed results of original and retrained models for Mol2vec and ChemBERTa on all datasets.

Table 26: Mol2vec results comparison on MoleculeNet classification tasks.

| Dataset | AUROC original | AUROC retrained | Gain |
|---|---|---|---|
| BACE | 81.07 | 78.86 | -2.21 |
| BBBP | 72.46 | 71.96 | -0.50 |
| HIV | 77.46 | 77.37 | -0.09 |
| ClinTox | 66.27 | 80.31 | +14.04 |
| MUV | 66.10 | 70.10 | +4.00 |
| SIDER | 67.61 | 66.79 | -0.82 |
| Tox21 | 77.49 | 76.82 | -0.67 |
| ToxCast | 65.27 | 65.83 | +0.56 |
| **Average** | 71.72 | 73.51 | **+1.79** |

Table 27: Mol2vec comparison on MoleculeNet regression tasks.

| Dataset | MAE original | MAE retrained | Gain |
|---|---|---|---|
| ESOL | 0.846 | 0.798 | -0.048 |
| FreeSolv | 2.876 | 2.327 | -0.549 |
| Lipophilicity | 0.709 | 0.697 | -0.012 |
| **Average** | 1.477 | 1.274 | **-0.203** |

Table 28: Mol2vec results comparison on TDC classification tasks.

| Dataset | AUROC original | AUROC retrained | Gain |
|---|---|---|---|
| ames | 80.57 | 81.06 | +0.49 |
| bioavailability_ma | 73.26 | 71.22 | -2.04 |
| cyp1a2_veith | 91.37 | 91.92 | +0.55 |
| cyp2c9_veith | 86.18 | 86.71 | +0.53 |
| cyp2c9_substrate_carbonmangels | 61.79 | 62.64 | +0.85 |
| cyp2c19_veith | 84.88 | 85.53 | +0.65 |
| cyp2d6_veith | 84.11 | 84.24 | +0.13 |
| cyp2d6_substrate_carbonmangels | 81.60 | 82.41 | +0.81 |
| cyp3a4_veith | 84.21 | 84.98 | +0.77 |
| cyp3a4_substrate_carbonmangels | 65.38 | 64.72 | -0.66 |
| dili | 91.76 | 95.07 | +3.31 |
| herg | 82.03 | 83.46 | +1.43 |
| herg_karim | 85.77 | 86.04 | +0.27 |
| hia_hou | 97.43 | 98.93 | +1.50 |
| pampa_ncats | 69.88 | 71.31 | +1.43 |
| pgp_broccatelli | 89.11 | 87.57 | -1.54 |
| sarscov2_3clpro_diamond | 68.92 | 70.22 | +1.30 |
| sarscov2_vitro_touret | 56.14 | 59.01 | +2.87 |
| **Average** | 79.69 | 80.39 | **+0.58** |

Table 29: Mol2vec results comparison on TDC regression tasks.

| Dataset | MAE original | MAE retrained | Gain |
|---|---|---|---|
| caco2_wang | 0.303 | 0.311 | 0.008 |
| clearance_hepatocyte_az | 36.953 | 37.425 | 0.472 |
| clearance_microsome_az | 28.449 | 30.415 | 1.966 |
| half_life_obach | 19.444 | 13.273 | -6.171 |
| ld50_zhu | 0.666 | 0.656 | -0.01 |
| ppbr_az | 9.393 | 9.552 | 0.159 |
| solubility_aqsoldb | 1.007 | 0.974 | -0.033 |
| vdss_lombardo | 4.119 | 3.762 | -0.357 |
| **Average** | 12.542 | 12.046 | **-0.496** |

Table 30: Mol2vec results comparison on WelQrate datasets.

| Dataset | Original | | Retrained | | Gain | |
|---|---|---|---|---|---|---|
| | AUROC | BEDROC | AUROC | BEDROC | AUROC | BEDROC |
| AID1798 | 63.12 | 20.35 | 66.46 | 20.81 | +3.34 | +0.46 |
| AID1843 | 76.89 | 47.47 | 76.97 | 49.35 | +0.08 | +1.88 |
| AID2258 | 70.12 | 32.51 | 73.17 | 37.25 | +3.05 | +4.74 |
| AID2689 | 74.40 | 46.48 | 73.97 | 43.94 | -0.43 | -2.54 |
| AID435008 | 69.52 | 35.83 | 73.83 | 39.44 | +4.31 | +3.61 |
| AID435034 | 70.48 | 22.52 | 75.07 | 25.59 | +4.59 | +3.07 |
| AID463087 | 86.24 | 43.37 | 86.23 | 42.26 | -0.01 | -1.11 |
| AID485290 | 75.09 | 35.87 | 76.58 | 37.73 | +1.49 | +1.86 |
| AID488997 | 73.09 | 36.00 | 72.44 | 37.72 | -0.65 | +1.72 |
| **Average** | 73.22 | 35.60 | 74.97 | 37.12 | **+1.75** | **+1.52** |

Table 31: Mol2vec results comparison on ToxBench dataset.

| Dataset | Metric | Original | Retrained | Gain |
|---------|--------|----------|-----------|------|
| ToxBench | Pearson correlation | 0.535 | 0.579 | 0.044 |
| | RMSE | 3.982 | 3.843 | -0.139 |

Table 32: Mol2vec results comparison on ApisTox dataset.

| Dataset | AUROC original | AUROC retrained | Gain |
|---------|----------------|-----------------|------|
| ApisTox | 70.34 | 72.57 | +2.23 |

Table 33: ChemBERTa results comparison on MoleculeNet classification tasks.

| Dataset | AUROC original | AUROC retrained | Gain |
|---------|----------------|-----------------|------|
| BACE | 71.52 | 73.80 | +2.28 |
| BBBP | 73.93 | 71.65 | -2.28 |
| HIV | 74.11 | 74.54 | +0.43 |
| ClinTox | 98.36 | 99.41 | +1.05 |
| MUV | 49.58 | 53.54 | +3.96 |
| SIDER | 59.58 | 61.86 | +2.28 |
| Tox21 | 68.77 | 69.81 | +1.04 |
| ToxCast | 59.20 | 59.15 | -0.05 |
| **Average** | 69.38 | 70.47 | **+1.09** |

Table 34: ChemBERTa comparison on MoleculeNet regression tasks.

| Dataset | MAE original | MAE retrained | Gain |
|---------|--------------|---------------|------|
| ESOL | 1.299 | 0.998 | -0.301 |
| FreeSolv | 2.478 | 2.169 | -0.309 |
| Lipophilicity | 0.760 | 0.759 | -0.001 |
| **Average** | 1.512 | 1.309 | **-0.200** |

Table 35: ChemBERTa results comparison on TDC classification tasks.

| Dataset | AUROC original | AUROC retrained | Gain |
|---------|----------------|-----------------|------|
| ames | 74.40 | 74.85 | +0.45 |
| bioavailability_ma | 67.94 | 68.17 | +0.23 |
| cyp1a2_veith | 87.38 | 87.38 | +0.00 |
| cyp2c9_veith | 81.75 | 83.87 | +2.12 |
| cyp2c9_substrate_carbonmangels | 62.38 | 63.99 | +1.61 |
| cyp2c19_veith | 81.79 | 83.16 | +1.37 |
| cyp2d6_veith | 77.96 | 80.64 | +2.68 |
| cyp2d6_substrate_carbonmangels | 74.84 | 78.46 | +3.62 |
| cyp3a4_veith | 80.51 | 82.45 | +1.94 |
| cyp3a4_substrate_carbonmangels | 62.62 | 59.67 | -2.95 |
| dili | 67.13 | 75.04 | +7.91 |
| herg | 68.42 | 72.37 | +3.95 |
| herg_karim | 79.69 | 81.82 | +2.13 |
| hia_hou | 81.34 | 88.35 | +7.01 |
| pampa_ncats | 64.34 | 69.29 | +4.95 |
| pgp_broccatelli | 83.54 | 82.46 | -1.08 |
| sarscov2_3clpro_diamond | 63.84 | 59.63 | -4.21 |
| sarscov2_vitro_touret | 59.36 | 50.25 | -9.11 |
| **Average** | 73.29 | 74.55 | **+1.26** |

Table 36: ChemBERTa results comparison on TDC regression tasks.

| Dataset | MAE original | MAE retrained | Gain |
|---|---|---|---|
| caco2_wang | 0.473 | 0.444 | -0.029 |
| clearance_hepatocyte_az | 37.492 | 38.493 | +1.001 |
| clearance_microsome_az | 30.402 | 30.289 | -0.113 |
| half_life_obach | 29.757 | 16.357 | -13.400 |
| ld50_zhu | 0.692 | 0.686 | -0.006 |
| ppbr_az | 10.446 | 10.567 | +0.121 |
| solubility_aqsoldb | 1.309 | 1.211 | -0.098 |
| vdss_lombardo | 3.524 | 4.709 | +1.185 |
| **Average** | 14.262 | 12.845 | **-1.417** |

Table 37: ChemBERTa results comparison on WelQrate datasets.

| Dataset | Original | | Retrained | | Gain | |
|---|---|---|---|---|---|---|
| | AUROC | BEDROC | AUROC | BEDROC | AUROC | BEDROC |
| AID1798 | 60.71 | 15.63 | 64.58 | 18.67 | +3.87 | +3.04 |
| AID1843 | 66.12 | 30.80 | 64.39 | 29.55 | -1.73 | -1.25 |
| AID2258 | 66.29 | 30.02 | 68.13 | 30.70 | +1.84 | +0.68 |
| AID2689 | 58.47 | 24.61 | 65.46 | 32.31 | +6.99 | +7.70 |
| AID435008 | 63.60 | 29.10 | 61.43 | 25.45 | -2.17 | -3.65 |
| AID435034 | 62.41 | 25.16 | 63.57 | 24.59 | +1.16 | -0.57 |
| AID463087 | 73.12 | 31.01 | 76.88 | 33.63 | +3.76 | +2.62 |
| AID485290 | 67.58 | 25.64 | 66.22 | 24.51 | -1.36 | -1.13 |
| AID488997 | 61.17 | 23.16 | 61.86 | 25.31 | +0.69 | +2.15 |
| **Average** | 64.39 | 26.13 | 65.99 | 28.26 | **+1.45** | **+1.07** |

Table 38: ChemBERTa results comparison on ToxBench dataset.

| Dataset | Metric | Original | Retrained | Gain |
|---|---|---|---|---|
| ToxBench | Pearson correlation | 0.448 | 0.447 | -0.001 |
| | RMSE | 4.241 | 4.224 | -0.017 |

Table 39: ChemBERTa results comparison on ApisTox dataset.

| Dataset | AUROC original | AUROC retrained | Gain |
|---|---|---|---|
| ApisTox | 71.65 | 68.84 | -2.81 |

