# OpenReview forum: "MolPILE - large-scale, diverse dataset for molecular representation learning"
_ICLR.cc/2026/Conference — ICLR 2026 Conference Withdrawn Submission_

### Official Review · Reviewer_A7Hp · 2025-10-28

**Soundness:** 2
**Presentation:** 2
**Contribution:** 2
**Rating:** 2
**Confidence:** 4

**Summary:**

The paper presents MolPILE, a large-scale dataset of 222 million compounds curated from six public chemical databases. The authors aim to provide a foundation dataset for molecular representation learning. The dataset construction pipeline includes preprocessing, standardization with a QSAR workflow, and physicochemical filtering, followed by evaluation through retraining Mol2vec and ChemBERTa-2 on MoleculeNet and other benchmarks.

**Strengths:**

- The paper is clearly written and well-organized, with comprehensive background research and thorough comparison to prior datasets.

- The analysis of elemental composition, molecular filters, and property distributions is detailed and informative.

- The motivation—to create a large unified dataset for molecular pretraining—is relevant and valuable for the field.

**Weaknesses:**

1. **Limited experimental scope:** The evaluation includes only two outdated SMILES-based baselines (Mol2vec and ChemBERTa-2). The absence of modern graph- or 3D-based models (e.g., GROVER, GraphMAE, UniMol) substantially weakens the claim that “MolPILE benefits molecular representation learning”, as the comparison lacks breadth and contemporary relevance.

2. **Minor presentation issues:** Figure 1, while conceptually clear, appears visually simplistic; incorporating schematic icons or stylized elements could enhance interpretability and visual appeal. In addition, Table 1 employs inconsistent numerical units (M/k), which should be standardized for clarity and professional presentation.

**Questions:**

1. **Feasibility filters appear overly permissive:**

The chosen thresholds (MW ≤ 2500, RB ≤ 60, logP ∈ [−10, 25]) seem inconsistent with the stated  focus on “small molecules”, as they may still include peptides, nucleic acids, polysaccharides, and other large or complex entities. The authors could provide quantitative justification—such as the percentage of entries containing peptide bonds, nucleic acid motifs, or polysaccharide fragments—to demonstrate exclusion.

Additionally, note that logP values in extreme ranges are themselves model-predicted and unreliable; the range [−10, 25] may not be chemically meaningful, as such out-of-distribution (OOD) predictions are typically inaccurate.

2. **Rationale for mixing heterogeneous data sources:**

The dataset combines large experimental databases,  screening catalogs, and natural product databases, which differ significantly in their chemical-space distributions and functional purposes. What is the motivation for merging these domains?

Why did the authors choose to operate at the level of “synthesizable compounds” rather than, for instance, integrating synthetic reaction–based datasets or restricting their scope to drug-like small molecules? Whereas many previous studies focus exclusively on natural products to maintain distributional consistency, what advantages or empirical evidence support the authors’ decision to adopt this scope?

3. **The reported experimental results raise multiple questions:**

    - The absolute performance of the models falls short of current state-of-the-art results.

    - In Table 26, 5 of 8 tasks show negative gains, with overall improvement seemingly driven mainly by ClinTox. Additionally, simply averaging the AUROC gains across different MoleculeNet tasks may not have clear statistical meaning.

    - Some “original” ChemBERTa scores (e.g., 98.36 AUROC on ClinTox in Table 33) diverge sharply from those in the original literature[1]. Could the authors clarify the evaluation setup, number of random seeds, and whether these discrepancies stem from preprocessing, data splits, or metric computation?

[1] Ahmad, Walid, et al. “ChemBERTa-2: Towards Chemical Foundation Models.” arXiv preprint arXiv:2209.01712, 2022

---

### Official Review · Reviewer_wkp7 · 2025-10-28

**Soundness:** 2
**Presentation:** 2
**Contribution:** 2
**Rating:** 4
**Confidence:** 3

**Summary:**

This paper presents MolPILE, a new large-scale dataset for molecular representation learning. The authors identify limitations in existing datasets, such as insufficient size, poor quality control, and reduced structural diversity due to excessive filtering. To address these issues, MolPILE compiles 220 million compounds through an automated pipeline, ensuring scale, diversity, and quality. Retraining existing models on MolPILE demonstrates improved generalization, highlighting the dataset's value.

**Strengths:**

- Clear Motivation: The paper convincingly outlines the limitations of current pretraining datasets and the need for a unified, large-scale chemical dataset. Constructing high-quality datasets is a fundamental step for broadening its application across diverse chemical domains.
- Systematic Data Construction: The multi-stage pipeline—preprocessing, standardization, and filtering—is applied. Employing only minimal and chemically valid filtering enables the construction of a dataset that is both larger in scale and structurally more diverse.

**Weaknesses:**

- Insufficient Validation of Dataset Quality: Although the paper provides various analyses on size, diversity, and synthesizability, these remain descriptive rather than evidential. It is unclear whether the reported statistics truly demonstrate that MolPILE is a high-quality dataset that improves model learning or generalization.
- Limited Evidence for Generalization: The evaluation covers only a narrow set of models and tasks, leaving it unclear whether MolPILE truly enhances transferability across different molecular learning frameworks.
- Experimental Design Concerns: Mol2vec and ChemBERTa embeddings were used as feature extractors and then fed into a random forest classifier for prediction. The evaluation setup does not clearly reflect common practices and may not accurately capture the true capability of the pretrained models.
- Missing Analysis on Dataset Scaling: MOLFILE provides subsets of different sizes, such as 1M, 5M, and 10M, to support scalable experiments. If the dataset is truly high-quality, it is necessary to verify whether performance improvements are consistently observed across different dataset sizes compared to existing datasets of similar scale.

**Questions:**

- Generalization Across Methods: Could the authors test MolPILE’s generalization effect by applying it to other molecular learning methods, using each method’s original experimental setup, to verify whether similar improvements are consistently observed?
- Element Group Analysis: Figure 2 shows only marginal differences in element distributions across datasets. Does this actually lead to broader chemical coverage or measurable performance gains?

---

### Official Review · Reviewer_Ngyd · 2025-10-31

**Soundness:** 3
**Presentation:** 3
**Contribution:** 2
**Rating:** 4
**Confidence:** 5

**Summary:**

This paper introduces MolPILE, a large-scale dataset of approximately 222M experimentally validated small molecules designed for molecular representation learning. The authors describe a standardized multi-stage curation pipeline for data cleaning, structure normalization, and feasibility filtering. They analyze MolPILE in terms of size, chemical diversity, and synthesizability, and demonstrate that retraining existing models (Mol2vec and ChemBERTa) on MolPILE leads to improved performance across a wide range of molecular property prediction benchmarks.

**Strengths:**

1. The work targets an important problem in molecular ML by proposing a standardized large-scale dataset.

2. The curation pipeline is clearly defined, reproducible, and appropriate for large-scale preprocessing.

3. Extensive dataset quality and diversity analyses are provided.

**Weaknesses:**

1. Effect of Data Scale vs. Data Curation is Not Clearly Disentangled.

The performance gains reported for Mol2vec and ChemBERTa may be influenced heavily by the scale difference rather than the data quality. The paper does not control for dataset size when comparing pretraining sources. For instance, recent versions of ZINC and PubChem offer significantly larger collections than the proposed dataset, yet the authors do not include experiments where MolPILE and alternative datasets are sampled to matched scales. As a result, it remains unclear whether the improvements stem from the dataset’s curation pipeline (diversity, structure normalization, feasibility filtering) or simply from training on more data.

2. Limited Model Coverage Restricts the Strength of Claims.

The experiments retrain only two relatively small and older architectures (Mol2vec and ChemBERTa). These models do not represent the current state of the field, which includes large graph transformers, equivariant 3D neural networks, etc. Without evaluating at least one contemporary large model, it is difficult to determine how useful MolPILE will be for modern molecular ML and foundation model scaling.

3. Vocabulary and Tokenization Limitations Not Investigated.

ChemBERTa trained on MolPILE shows degraded performance on ApisTox, attributed to tokenization limits (vocabulary not covering rare structural patterns). However, the authors do not evaluate alternative tokenization strategies (e.g., SMILES BPE scaling, SELFIES, fragment tokenization), nor do they quantify how vocabulary coverage correlates with downstream performance. This leaves unclear how well MolPILE supports SMILES language modeling when chemical space becomes highly diverse.

4. No Ablation on Filtering Pipeline.

While the dataset construction pipeline includes multiple important steps (standardization, feasibility filtering, deduplication), the paper does not study how each step contributes to data quality or downstream model performance. An ablation analysis could reveal, for example, whether feasibility filtering meaningfully improves representation learning, or whether diversity selection is the primary factor.

**Questions:**

1. Can the authors perform controlled experiments where MolPILE is downsampled to the same size as baseline datasets (e.g., ZINC subsets), in order to demonstrate that performance improvements arise from the curation pipeline rather than dataset scale alone?

2. How would MolPILE perform when evaluating MolPILE-pretrained models on 3D-aware benchmarks or reaction prediction tasks?

---

### Official Review · Reviewer_uEqp · 2025-11-06

**Soundness:** 3
**Presentation:** 3
**Contribution:** 3
**Rating:** 4
**Confidence:** 4

**Summary:**

This paper introduces MolPILE, a large-scale, diverse, and rigorously curated dataset for molecular representation learning, developed to overcome the size, diversity, and quality limitations of existing small-molecule datasets.

**Strengths:**

1. With 222 million compounds, MolPILE achieves a far larger scale than commonly used datasets like ChEMBL (2.4M) and ZINC (12.7M), providing sufficient data volume for training large-scale foundation models
2. Second, it exhibits superior structural and molecular type coverage—covering halogens, metalloids, metals (unlike ZINC and GDB-17, which lack metals/metalloids), over 3.6 million unique Bemis-Murcko scaffolds, and 1.09 million salts, effectively expanding the chemical space for model pretraining
3. The authors validate the dataset’s value by retraining 1D-based models (Mol2vec and ChemBERTa) on MolPILE, showing consistent performance gains across 48 benchmarks, demonstrating its effectiveness for 1D molecular representation learning

**Weaknesses:**

1. MolPILE only provides 2D molecular representations (SMILES/InChI) without 3D conformations, restricting its applicability to 3D-based pretraining methods or multimodal fusion approaches; the validation is solely focused on 1D models, failing to support broader molecular learning paradigms.

2. While the dataset claims high diversity (e.g., metal-containing compounds, improved structural diversity with ~3% normalized #Circles), there is a lack of targeted validation on downstream tasks specific to these diverse subsets—for example, no dedicated experiments on agrochemistry tasks (despite covering metals) or natural product property prediction (despite including SuperNatural3/COCONUT data) to confirm if the enhanced diversity translates to task-specific improvements.

3. Although scale is a core highlight, the paper lacks detailed ablation studies on molecular data scaling laws. It does not systematically demonstrate whether increasing data volume leads to stepwise improvements in molecular property prediction accuracy for classical pretraining methods, which is essential to justify the necessity of such a large-scale dataset.

**Questions:**

See weakness

---

### Author Response · Authors · 2025-11-23

In the light of, frankly, unreasonable reviews misrepresenting the nature and core contributions of the paper, we decided to write a single answer to the future readers of this submission. Seeing the abysmal quality of the reviews, both for our submission and for ICLR in general this year, we lost faith in the quality of this conference, and do not seek publication there. We leave this response for future readers, in order to fully appreciate the nonsensical nature of the reviews provided.

In short, fulfilling the review requirements would amount to writing a full PhD thesis on the topic of molecular representation learning models. All reviewers request more content, more experiments, more models, more research. We note that MolPILE is a dataset, submitted under Datasets & Benchmarks category, focusing on the data and its analysis. Despite this, all reviewers expect basically a whole thesis researching molecular scalability, neural scaling laws, and training a host of chemical LLMs. Further, reviews contain errors directly contradicting the content of the paper. While we do not claim they are necessarily LLM-generated, the combination of a) obvious mistakes contradicting the paper, and b) generic requests of "more this, more that" indicate high probability of generated reviews or surface-level skimming through our work, rather than proper reviewing process.

Summarizing the most unreasonable requests from reviewers:
1. Reviewers uEqp & wkp7- "paper lacks detailed ablation studies on molecular data scaling laws", "Missing Analysis on Dataset Scaling". This literally requests retraining a large series of models, probably with different model types and modalities (e.g. SMILES transformers vs GNNs). This topic in itself is easily large enough for another paper.
2. Reviewer Ngyd - "Limited Model Coverage Restricts the Strength of Claims". Reviewer expects us to retrain graph transformers, equivariant 3D neural networks (despite MolPILE explicitly not containing 3D structures), or more models. This topic would be enough for a full series of papers in the future. Further, we point out that analysis of Praski et al. (https://arxiv.org/abs/2508.06199) explicitly lists ChemBERTa as one of the best models. Older models do not mean worse in chemistry.
3. Reviewer Ngyd - "Vocabulary and Tokenization Limitations Not Investigated". Reviewer expects us to retrain more transformers, try out different tokenization schemes, perform vocabulary size ablation etc. This is, again, material for a full separate paper.
4. Reviewer A7Hp - "Why did the authors choose to operate at the level of “synthesizable compounds” rather than, for instance, integrating synthetic reaction–based datasets or restricting their scope to drug-like small molecule". Because this is literally the aim of the paper, and its entirety explains it. If someone expects a completely different dataset, then they can create it and write a paper on that. Changing this would literally require throwing out our whole work and misses the point of MolPILE.

In the light of those numerous problems, we withdraw our submission, and will submit the manuscript to a properly professional journal.

---

> ### Author Response · Authors · 2025-11-23
>
> Summarizing review errors:
> 1. Reviewer uEqp - "there is a lack of targeted validation on downstream tasks specific to these diverse subsets—for example, no dedicated experiments on agrochemistry tasks". ApisTox is exactly the agrochemistry dataset, covering ecotoxicology area, core part of novel agrochemical design.
> 2. Reviewer wkp7 - "Insufficient Validation of Dataset Quality". Our paper contains so many analyses of dataset quality that we explicitly created a section 4.5 Further analyses, pointing readers to 16 pages of appendices. We provided the most comprehensive analyses of molecular datasets quality to date.
> 3. Reviewer wkp7 - "Limited Evidence for Generalization. The evaluation covers only a narrow set of models and tasks". We cover 48 datasets, and we are not aware of any other work in this area that provides such a comprehensive evaluation. We train two representative models, as this is datasets & benchmarks paper.
> 4. Reviewer wkp7 - "evaluation setup does not clearly reflect common practices". There is literally no other way to use Mol2Vec than we did, i.e. as a feature extractor. The original authors did exactly the same thing. Further, we followed the protocol of Praski et al. (https://arxiv.org/abs/2508.06199), which is cited in the corresponding section.
> 5. Reviewer A7Hp - "Feasibility filters appear overly permissive". Appendix D contains 3 whole pages with detailed motivation and statistics.
> 6. Reviewer A7Hp - "The absolute performance of the models falls short of current state-of-the-art results". Yes, because it was never the point. We care only if MolPILE improves results with the given approach, not if it gives the absolute best results.
> 7. Reviewer A7Hp - "Some “original” ChemBERTa scores (e.g., 98.36 AUROC on ClinTox in Table 33) diverge sharply from those in the original literature". This is explained in detail in Section 5, where we explicitly state that original model means original model, not original scores from the paper. We use embeddings + Random Forest, not finetuning, so don't expect the same results. What we show is the performance difference under the unified protocol.
> 8. Reviewer A7Hp - "Could the authors clarify the evaluation setup, number of random seeds, and whether these discrepancies stem from preprocessing, data splits, or metric computation?". Already covered in Section 5 and Appendix M.
> 9. Reviewer A7Hp - "The evaluation includes only two outdated SMILES-based baselines (Mol2vec and ChemBERTa-2).". Both of those "outdated" baselines outperform all alternatives in recent analysis of Praski et al. (https://arxiv.org/abs/2508.06199), including GROVER and UniMol mentioned by reviewer, by a large margin. Further, ChemBERTa-2 is from 2022, and then reviewer proceeds to list 3 models older or from the same time: GROVER (2020), GraphMAE (2022), UniMol (ArXiv 2022, ICLR 2023).

---

### Note · Authors · 2025-11-23

I have read and agree with the venue's withdrawal policy on behalf of myself and my co-authors.